# Structural basis for the bi-specificity of USP25 and USP28 inhibitors

Jonathan Vincent Patzke [ID] [1,5], Florian Sauer [ID] [1,5], Radhika Karal Nair [1,5], Erik Endres [ID] [2],
Ewgenij Proschak [3,4], Victor Hernandez-Olmos [4], Christoph Sotriffer [ID] [2] & Caroline Kisker [ID] [1✉]

## Abstract

**The development of cancer therapeutics is often hindered by the fact that specific oncogenes cannot be directly pharmaceutically addressed. Targeting deubiquitylases that stabilize these oncogenes provides a promising alternative. USP28 and USP25 have been identified as such target deubiquitylases, and several small-molecule inhibitors indiscriminately inhibiting both enzymes have been developed. To obtain insights into their mode of inhibition, we structurally and functionally characterized USP28 in the presence of the three different inhibitors AZ1, Vismodegib and FT206. The compounds bind into a common pocket acting as a molecular sink. Our analysis provides an explanation why the two enzymes are inhibited with similar potency while other deubiquitylases are not affected. Furthermore, a key glutamate residue at position 366/373 in USP28/USP25 plays a central structural role for pocket stability and thereby for inhibition and activity. Obstructing the inhibitor-binding pocket by mutation of this glutamate may provide a tool to accelerate future drug development efforts for selective inhibitors of either USP28 or USP25 targeting distinct binding pockets.**

**Keywords** Cancer; DUB; Inhibitor; USP25; USP28
**Subject Categories** Pharmacology & Drug Discovery; Post-translational Modifications & Proteolysis; Structural Biology

## Introduction

Deubiquitylases (DUBs) catalyze the removal of Ubiquitin (Ub) from a substrate, most commonly by cleaving the isopeptide bond between the Ub C-terminus and the ε-amino group of a lysine on another Ub molecule or the substrate protein. Depending on their substrate specificity, this leads to the suppression or complete reversal of the effects of ubiquitylation, thus rendering DUBs critical players in regulating most cellular processes, such as protein degradation, localization, and control of protein activity in

eukaryotes (Swatek and Komander, 2016; Clague et al, 2019). The altered expression or function of DUBs is often associated with the onset and progression of various diseases. Ubiquitin-specific proteases (USPs) are cysteine proteases, which constitute the largest of the seven currently known human DUB families. They have become the center of interest as potentially well-druggable targets for pharmacological therapies, as many regulate essential pathways and display increased activity in a range of cancerous and non-cancerous malignancies. However, their close evolutionary relationship, combined with a broad spectrum of biological functions, represents a serious challenge for the design of small-molecule inhibitors. Drug screening campaigns have therefore resulted in bi- or poly-selective molecules for several DUBs, thus potentially lacking the required selectivity for pharmacological use (Aleo et al, 2006; Ye et al, 2009; Kapuria et al, 2010; Altun et al, 2011; Weinstock et al, 2012; Issaenko and Amerik, 2012; D'Arcy et al, 2015).

USP28 and USP25 are two closely related members of the USP family, sharing an identical domain architecture and a high sequence similarity, being most pronounced in their central catalytic domains with an identity of 57% (Sauer et al, 2019). Despite these commonalities, they fulfill distinct biological roles and differ significantly in their catalytic activities.

USP28 acts as a regulator of cell proliferation and differentiation (Wu et al, 2013; Zhao et al, 2019), is involved in DNA damage repair (Knobel et al, 2014), genome maintenance (Wang et al, 2021b), and apoptosis (Müller et al, 2020) by its deubiquitylation activity targeting specific nuclear proteins, thus preventing their degradation by the proteasome. Several of USP28's substrates, such as the transcription factors c-Myc, HIF1α, and ΔNp63, the lysine-specific demethylase LSD1 and the helicase RecQ5, act as oncogenes in a variety of cancers rendering USP28 itself an essential oncogene (Popov et al, 2007; Wu et al, 2013; Du et al, 2019; Prieto-Garcia et al, 2020; Wang et al, 2021b). However, in a melanoma model, it has also been found to stabilize p53, suggesting that USP28 additionally bears functions as a tumor suppressor (Müller et al, 2020). Like USP28, USP25 is considered a tumor promoter. It stabilizes tankyrases and thereby acts as a positive regulator of Wnt-signaling (Xu et al, 2017). Furthermore, it plays an essential role in the metabolic reprogramming in cells of the

[1]Rudolf Virchow Center for Integrative and Translational Bioimaging, Institute for Structural Biology, Julius-Maximilians-University Würzburg, Würzburg, Germany. [2]Institute of Pharmacy and Food Chemistry, Julius-Maximilians-University Würzburg, Würzburg, Germany. [3]Institute of Pharmaceutical Chemistry, Goethe-University, Frankfurt am Main, Germany. [4]Fraunhofer Institute for Translational Medicine and Pharmacology ITMP, Frankfurt am Main, Germany. [5]These authors contributed equally: Jonathan Vincent Patzke, Florian Sauer, Radhika Karal Nair. ✉E-mail: caroline.kisker@virchow.uni-wuerzburg.de

pancreatic cancer tumor core through the stabilization of the HIF1α transcription factor (Nelson et al, 2022). Next to this, USP25 is involved in a broad spectrum of cell-type and tissue-specific cellular pathways, including insulin-dependent glucose metabolism (Habtemichael et al, 2018), the positive and negative regulation of immune cell activity (Zhong et al, 2012, 2013; Lin et al, 2015), and ER-associated protein folding (Blount et al, 2012).

Previous studies by our group and others showed USP28 to be constitutively active, while USP25 is auto-inhibited (Liu et al, 2018; Sauer et al, 2019; Gersch et al, 2019). Crystal structures of the catalytic domains of both proteins provided an explanation for this difference in activity which originates from an oligomerization domain (UCID) embedded into the core-USP domain. In USP28, the UCID is responsible for the formation of a constitutively active dimer. In USP25, however, a large loop (UCID-tip) anchored into a cleft in the S1-binding site interlinks two dimers to form an auto-inhibited tetramer and dissociation into dimers is required for the DUB to reach full activity.

Recent high-throughput screening efforts have led to the discovery of a substantial number of relatively potent small-molecule inhibitors targeting USP28 activity (Wrigley et al, 2017; Wang et al, 2021a; Liu et al, 2020; Varca et al, 2021; Ruiz et al, 2021; Xu et al, 2023). However, where tested, all currently known compounds have also been found to inhibit USP25 with similar potency. Of these, three compounds have, since their discovery, been successfully used in different contexts for the treatment of cancer in humans or experimental animals.

AZ1 is one of three benzylaminoethanol derivatives (AZ1, AZ2, and AZ4) identified as potent inhibitors of USP28 and USP25 (Wrigley et al, 2017). It has been used as an in vivo probe to assess the function of both DUBs. Prieto-Garcia et al demonstrated that AZ1 blocks USP28-dependent stabilization of ΔNp63, leading to a decrease in SCC transplanted tumor size and number (Prieto-Garcia et al, 2020). AZ1 also mitigates the neuropathological hallmarks of Alzheimer's disease (AD) by attenuating microglial activation, of which USP25 is a critical regulator, as demonstrated in 5xFAD mice, a model for neuronal deficits linked to trisomy 21 (Zheng et al, 2021). Furthermore, inhibition of USP25 by AZ1 in a mouse model was shown to suppress inflammation linked to bacterial infections in the intestine and enhance the immune response while inhibiting the activity of USP25 in promoting intestinal cancer (Wang et al, 2020).

Vismodegib (Erivedge®, VSM) is an FDA-approved drug to treat unresectable or metastatic basal cell carcinoma (BCC) (Dlugosz et al, 2012). It is a competitive antagonist of the smoothened receptor (SMO) and therefore acts as an inhibitor of the hedgehog signaling pathway. In a recent study by H. Wang and colleagues (Wang et al, 2021a), VSM was shown to bind and inhibit USP28 and USP25. In cellulo studies using colorectal cancer cell lines showed that exposure to the drug blocked Ub-substrate binding to USP28 and USP25 and induced cytotoxic effects in a concentration-dependent and hedgehog-pathway-independent manner.

FT206 is a representative of a large (>200), patented compound series of substituted thienopyridine-carboxamides of USP28/USP25-bispecific inhibitors and the result of scaffold optimization efforts towards improved drug-metabolism and pharmacokinetic properties (Ruiz et al, 2021). The compound was found to exhibit cytotoxic effects in a USP28-dependent manner in lung SCC cells,

with a significantly higher efficiency than that observed for AZ1. FT206 also displayed cytotoxicity in mouse implanted USP25 dependent PDAC tumors. Here, exposure to FT206 induced the loss of the HIF1α transcription factor, thus preventing metabolic reprogramming and inducing apoptosis in the hypoxic tumor core (Nelson et al, 2022).

To address inhibitor binding and guide future development of a selective inhibitor against USP28 or USP25, we solved the high-resolution crystal structures of the USP28-inhibitor complexes comprising AZ1, VSM, and FT206, respectively. We observed that the three molecules bind at different positions into the same cleft formed between the USP-thumb and palm at the catalytic center distal portion of the S1-site, which is highly conserved between the two homologs. This cleft can also be observed in apo USP28 and the auto-inhibited USP25, in the latter it accommodates the auto-inhibitory UCID-tip. Inhibitor binding prevents conformational changes of the S1-site and thus its transition into a substrate-binding competent state. Via mutational probing, we validated the binding site in USP28, showed that the same cleft is targeted in USP25, and identified key residues responsible for inhibition.

## Results

### Structures of the USP28-inhibitor complexes

Initial attempts to obtain USP28-inhibitor co-crystal structures by soaking the inhibitors into crystals of the full-length USP28 catalytic domain (amino acids 149–707) (Sauer et al, 2019) were unsuccessful. Crystallization screening using different truncated variants of the USP28 catalytic domain led to two new crystal forms. USP28Δtip, a dimeric variant lacking the large disordered UCID "tip-region" (amino acids 149-458-SGSG-529-707), crystal-lized in space group $P2_122_1$ with two protein molecules in the asymmetric unit (ASU) and diffracted anisotropically to a resolution of 2.8–3.0 Å. USP28ΔUCID, a variant reduced to the core-USP domain by truncation of the entire UCID-subdomain (amino acids 149-399-GSGSGS-580-698), similar to a variant previously reported by the Komander lab (Gersch et al, 2019), crystallized in the rhombohedral space group $H3_2$, with one protein molecule per ASU. These crystals diffracted to a resolution of ~2.5 Å. Both crystal forms contained a high solvent fraction of ~70%, with a large portion of the S1 Ub-binding sites being solvent-exposed. The crystals were stable for an extended time in up to 3% DMSO and therefore used for inhibitor soaking.

The USP28Δtip variant was used to solve the inhibitor-bound structures containing AZ1 and FT206, respectively, with diffraction data extending to a resolution of 2.8 Å. The VSM-containing structure was obtained in complex with USP28ΔUCID using diffraction data to a resolution of 2.6 Å (Table 1). Well-defined electron density allowing the modeling of the entire inhibitor molecule was obtained after molecular replacement in the USP28ΔUCID molecule for VSM, in both USP28Δtip molecules of the asymmetric unit for AZ1 and in one USP28Δtip molecule for FT206 (molecule A), whereas the inhibitor was not completely defined in the second molecule (Appendix Fig. S1A). Next to low overall occupancy, the weak electron density may have originated from the simultaneous binding of (R-) and (S-) stereoisomers to molecule B due to the usage of a racemic mixture of FT206. A

**Table 1. Data collection and refinement statistics.**

| | USP28cat (ΔUCID) apo | USP28cat (ΔUCID) VSM | USP28cat (Δtip) P280H AZ1 | USP28cat (Δtip) FT206 |
|---|---|---|---|---|
| PDB ID | 8P19 | 8P14 | 8P1P | 8P1Q |
| **Data collection**[a] | | | | |
| Beamline | ESRF ID23-2 | ESRF ID23-2 | BESSY BL14.1 | EMBL P14 |
| Resolution range (Å) | 47.91–2.45 (2.55–2.45) | 47.97–2.57 (2.68–2.57) | 48.35–2.76 (3.01–2.76) | 46.08–2.76 (3.04–2.76) |
| Space group | $H3_2$ | $H3_2$ | $P2_122_1$ | $P2_122_1$ |
| Unit cell dimensions: | | | | |
| a, b, c (Å) | 106.63, 106.63, 327.67 | 106.63, 106.63, 329.85 | 100.44, 105.54, 178.71 | 100.44, 105.54, 178.71 |
| α, β, γ (°) | 90, 90, 120 | 90, 90, 120 | 90, 90, 90 | 90, 90, 90 |
| Observed reflections | 519,735 (60,001) | 448,422 (56,011) | 459,151 (23,917) | 415,432 (34,444) |
| Unique reflections | 26,861 (3008) | 23,472 (2825) | 34,461 (1723) | 34,993 (2917) |
| $R_{merge}$ | 0.17 (3.96) | 0.17 (4.12) | 0.24 (2.86) | 0.14 (2.16) |
| $R_{meas}$ | 0.18 (4.17) | 0.17 (4.23) | 0.27 (2.97) | 0.14 (2.26) |
| CC1/2 | 0.999 (0.369) | 0.999 (0.402) | 0.998 (0.393) | 0.998 (0.480) |
| Redundancy | 19.3 (19.9) | 19.1 (19.8) | 13.3 (13.9) | 11.9 (11.8) |
| $<I/\sigma I>$ | 13.7 (0.9) | 13.5 (0.9) | 10.9 (1.0) | 12.8 (1.2) |
| Completeness (%): | | | | |
| Spherical | 100 (100) | 100 (100) | 69.0 (15.2) | 71.0 (23.4) |
| Ellipsoidal | – | – | 94.3 (69.9) | 94.6 (76.8) |
| **Refinement** | | | | |
| Protein molecules/ASU | 1 | 1 | 2 | 2 |
| Resolution range (Å) | 47.97–2.45 | 44.46–2.57 | 48.35–2.76 | 46.09–2.79 |
| Reflections all/ free R set | 26,852/ 1201 | 23,464/ 1982 | 34,413/ 1213 | 34,606/ 1216 |
| $R_{work}/R_{free}$ (%) | 19.4/22.2 | 19.3/21.8 | 21.3/24.1 | 21.2/23.6 |
| No. of atoms: | | | | |
| Protein | 2789 | 2690 | 7104 | 6655 |
| Water | 102 | 89 | 47 | 55 |
| Non-water solvent | 8 | 4 | 5 | 8 |
| Inhibitor | – | 27 | 50 | 64 |
| Mean B-factors (Å²): | | | | |
| Protein | 88.62 | 93.74 | 76.29 | 101.48 |
| Water | 72.97 | 76.17 | 44.13 | 64.35 |
| Non-water solvent | 83.77 | 101.04 | 48.03 | 75.32 |
| Inhibitor | – | 73.52 | 62.50 | 96.69 |

**Table 1.** (continued)

| | USP28cat (ΔUCID) apo | USP28cat (ΔUCID) VSM | USP28cat (Δtip) P280H AZ1 | USP28cat (Δtip) FT206 |
|---|---|---|---|---|
| RMSD: | | | | |
| Bond lengths (Å) | 0.002 | 0.005 | 0.003 | 0.002 |
| Bond angles (°) | 0.477 | 0.811 | 0.558 | 0.527 |
| Ramachandran statistics: favored/ allowed/outliers (%) | 95.7/4.3/ 0.0 | 96.9/3.1/ 0.0 | 96.2/3.7/ 0.1 | 97.2/2.7/ 0.1 |

[a]Numbers in parentheses correspond to the highest resolution shells.

mutation of the USP28Δtip variant, P280H, was identified during structure refinement for the AZ1 bound structure. This residue is located within the helix α5 and α6 connecting loop and is not involved in any interactions with the inhibitor. To clarify whether this change influenced the activity or overall fold of the protein, the mutation was reversed to obtain the wild-type protein. An activity assay with the synthetic mono-Ub-substrate (Ub-Rho110) showed that the variant and the wild-type protein exhibit the same activity (Appendix Fig. S1B).

## Inhibitor binding

AZ1, VSM, and FT206 bind to USP28 at different overlapping positions inside the same hydrophobic cleft, located at the intersection of the USP-thumb and palm subdomains in the center of the concave part of the USP S1-site, which acts as a binding surface for the globular domain of the distal Ub moiety of the cleaved substrate (Figs. 1A and EV1A). This cleft is formed by residues of helices α1, α2, α5, and α6 on the USP-thumb and strands β8, β12, β13, and β16 on the USP-palm subdomain (Fig. 1A) and is equivalent to the cleft previously identified in USP25 as the binding site for the auto-inhibitory UCID-tip (Liu et al, 2018; Sauer et al, 2019; Gersch et al, 2019). A similar binding site has previously been identified for different USP7 inhibitors (Fig. EV1B) (Di Lello et al, 2017; Kategaya et al, 2017).

AZ1 binds to USP28 at the Ub-tail binding channel—distal end of the cleft, at the intersection between the thumb, palm, and finger subdomains. Its aromatic rings are oriented approximately perpendicular to each other and entirely buried in the cleft, while the aminoethanol chain extends towards the protein surface where it forms H-bonds with D265 and Q315 (Figs. 1B and EV1A). A major driver for the binding to USP28 appears to be the bromophenyl-moiety of the molecule, which is embedded parallel to helix α6 into a hydrophobic pocket mainly formed by the side chains of L180, F186, L264, M288 and F292 (Fig. EV1C; Appendix Fig. S1C). The aromatic F292 side chain interacts with the bromophenyl ring via π-stacking, thus providing additional stability to the interaction. Previous structure-activity analyses of the AZ-series of inhibitors showed that fluorine-carrying substitutions on the benzoxy-substituent of the AZ-scaffold are essential for USP28 binding and inhibition. Differently substituted molecules (AZ1: -F/-CF₃; AZ2: -O-CF₃; AZ4: 3x-F) inhibited USP28/USP25 in a high nanomolar to low micromolar range, while potency was

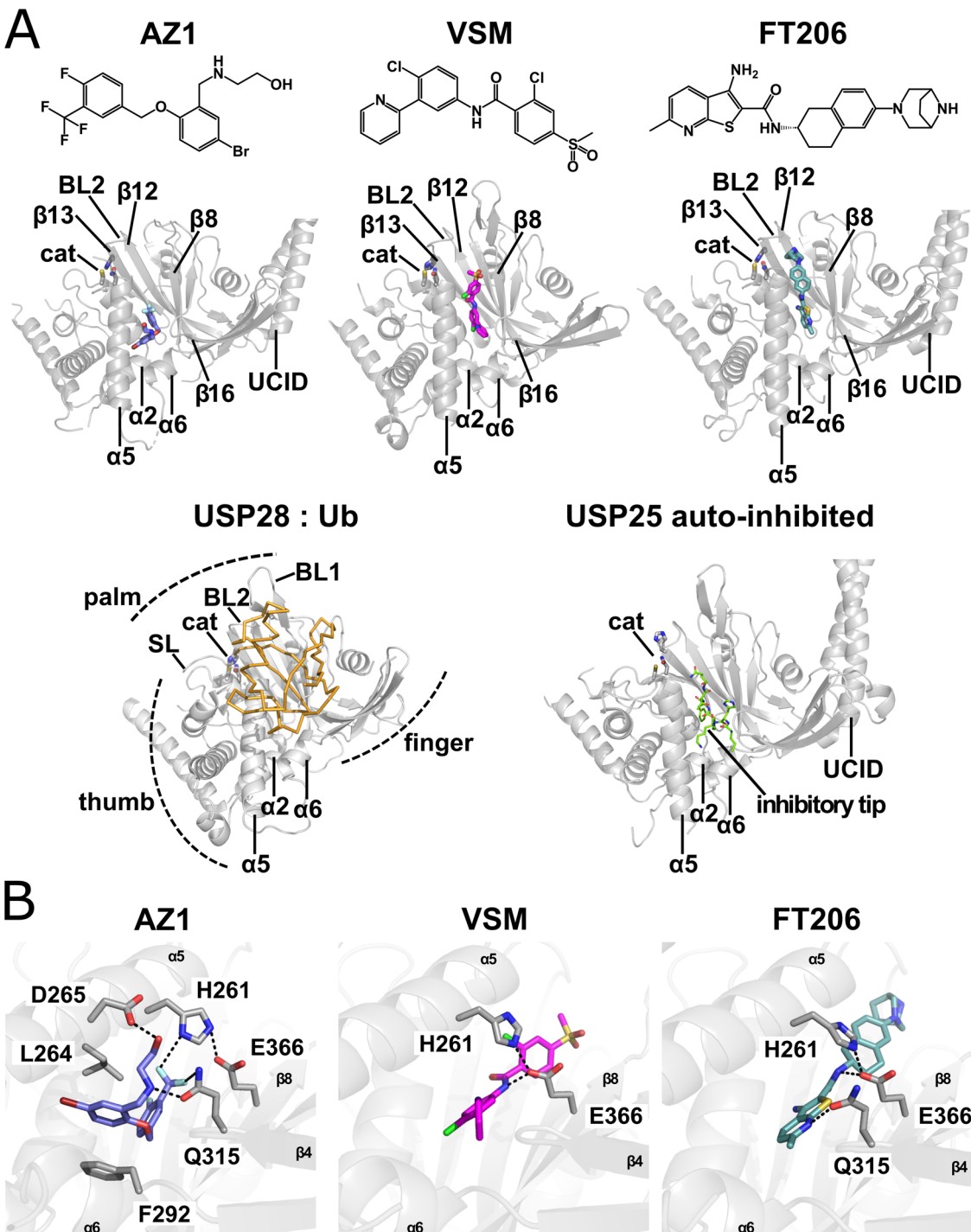

significantly diminished for the non-fluoro-substituted variant AZ3 (Wrigley et al, 2017).

The structure explains the potency-promoting effect of these substitutions. While the AZ1 fluorine in para position points towards the bottom of the binding cleft and is not involved in direct contacts, the trifluoro-methyl group in meta-position is involved in several non-covalent interactions with the surrounding residues. It approaches Cγ of the Q315 side chain amide at 3.6 Å, and Nδ of

H261 at 3.7 Å, which is in agreement with documented distances for fluorine-mediated multipolar interactions (Bissantz et al, 2010). At the same time the H261 side chain points towards E366, to form an arc reaching over the cleft, thereby locking AZ1 in its position leading to a closure of the binding site towards the protein surface (Fig. 1B). Computational protonation state analysis with PropKa (Søndergaard et al, 2011) suggests that the proximity of both side chains leads to a doubly protonated H261 thereby rendering

**Figure 1.   Crystal structures of USP28-inhibitor complexes.**

(A) Top panel: Structural formulae of AZ1 (left), VSM (center) and FT206 (right). Middle panel: The inhibitor binding cleft in USP28. AZ1 (left, blue), VSM (center, magenta) and FT206 (right, teal) bind at different positions of the same binding cleft, formed by the USP28 palm (β8, β12, β13, and β16) and thumb-helices (α1, α2, α5, and α6). The AZ1 binding site is located at the center of the S1-site, while the binding sites for VSM and FT206 are positioned further towards the Ub-tail binding channel. Bottom left: For comparison, the structure of USP28ΔUCID (PDB 6HEI; gray, cartoon) in complex with Ub-PA bound to the S1-site (orange, Cα-trace) is shown (Gersch et al, 2019; Data ref: Gersch and Komander, (2019a). Subdomains and regulatory elements of the catalytic/Ub-tail binding channel (cat: catalytic triad C171, H600, N617; SL: switching loop; BL1/BL2: blocking loops 1/2) are marked. Note that α1 is hardly visible and therefore not labeled. Its N-terminus is marked by the position of the catalytic cysteine. Bottom right: Crystal structure of the auto-inhibited USP25 catalytic domain (PDB 6H4J) (Sauer et al, 2019; Data ref: Klemm et al, 2019a). The same cleft accommodates the auto-inhibitory tip in the USP28 homolog USP25. Cleft-binding residues of the USP25-tip (I518–Q524) are shown as green sticks. (B) Closeup view of the inhibitor-binding sites. The side chains of residues involved in hydrogen bond formation with the inhibitors, L264 and F292, which move apart to facilitate the binding of AZ1 via π-stacking, and H261 and E366, which close the binding site towards the protein surface, are depicted as sticks. Hydrogen bonds between the inhibitors and USP28 and between H261 and E366, as well as multipolar contacts of the trifluoro-methyl group of AZ1 with adjacent residues are shown as dashed lines. Source data are available online for this figure.

engagement in multiple bonds with E366 and with the AZ1 trifluoro-methyl group possible.

VSM and FT206 bind to USP28 at similar sites located further towards the wider and open Ub-tail proximal position of the USP28-inhibitor-binding cleft (Fig. 1A,B). This is due to the rigidity of their central linker amides, which enforce a more extended conformation compared to AZ1. Consequently, VSM's central chlorobenzene and FT206's amino-methyl-thienopyridine moieties are oriented parallel to α5 and overlap with each other in the position of the AZ1 "outer" fluoro-substituted benzyl-moiety at the center of the binding cleft (Fig. EV1A,C; Appendix Fig. S1C). This places the amide bond nitrogens of both molecules below the H261–E366 arc, where they form hydrogen bonds with E366's carboxyl group. VSM's pyridine ring in meta-position of the central ring reaches towards the protein surface by filling the gap between Q315 and L264, while the chlorobenzene-methylsulfone moiety is oriented along the wider end of the binding cleft, thus reaching the N-terminal end of α5. Similarly, FT206's diaza-bicyclo-octane moiety extends into the Ub-tail binding site, where it nearly reaches blocking loop 2 (BL2). Despite the availability of several other potential hydrogen bond donor and acceptor sites on VSM's and on FT206's heterocycles, only one further hydrogen bond between Q315 and the pyridine nitrogen in FT206 is formed and no additional hydrogen bond between VSM and USP28. The remainder of the interaction for both molecules is mediated by van-der-Waals contacts. The observation that only a small number of specific hydrogen bonds are required for USP28 binding, is further supported by the substantial number of high potency USP28/USP25 inhibitors, that are like FT206 based on the thienopyridine-carboxamide scaffold, but differ substantially in size, chemical composition and number of their substitutions (Guerin et al, 2020).

## Mechanism of inhibition

Previous data suggested that AZ1 and VSM act as allosteric inhibitors of USP28. Kinetic analysis of full-length USP28 inhibition by AZ1 revealed a non-competitive mode of inhibition (Wrigley et al, 2017). Hydrogen deuterium exchange mass spectrometry (HDX-MS) experiments of USP28's binding site for VSM did not only point to the binding cleft-lining USP28 helices, but also to significant changes in proton-deuteron exchange rates in helices α3 and α4 on the outer surface of the thumb and at the far

side of the cleft bearing S1-site, indicating structural changes on the thumb upon inhibitor binding (Wang et al, 2021a).

Our inhibitor-bound structures reveal that all three examined small molecules inhibit USP28 by a common allosteric mechanism. Comparison of here and previously solved apo and Ub-bound structures of USP28 shows that upon Ub-binding to the S1-site, the catalytic domain undergoes a structural transition from an "open" apo form to a more closed substrate-bound state with major conformational changes in the thumb domain (Sauer et al, 2019; Gersch et al, 2019). Motions of the thumb-core helices α3, α4, and α5 remodel the S1-site to accommodate and bind Ub (Fig. 2A). Helix α5 moves slightly away from the catalytic channel and bends its N-terminal portion from an "upward" apo-position to a "downward" position, towards the palm β-sheet. This leads to a narrowing of the gap between the thumb and the palm, thereby reshaping the concave portion of the S1-site, responsible for the formation of a major fraction of direct bonds with the substrate mediated by residues along α5, and an increase in distance between the α5 N-terminus and blocking loops 1/2 (Figs. 2A and EV2A). As a result, a widening of the catalytic channel is achieved which permits binding of the Ub C-terminal tail, and the side chains of D255 and E258 are placed in a position where they can interact with Ub R72 and L73, respectively. Simultaneously, α5 movement is indirectly relayed via α3, to move α4 away from the catalytic channel. This increases the distance between the α4 C- and α5 N-termini, thus placing the helix-connecting switching loop (SL) in a position where it can be placed onto the Ub-tail and guide its C-terminus into the catalytic center.

AZ1, VSM, and FT206 inhibit the enzyme by locking the USP-thumb in an apo-like position due to their binding between α5 and the palm, where they act as wedge to prevent thumb transition into a substrate-binding competent state (Figs. 2B and EV2A/B).

Depending on the location of the inhibitor-binding site, helix α5 is locked in different positions, leading to distinct alterations of the Ub globular domain binding- and catalytic channel parts of the S1-site. The placement of VSM's chlorobenzene-methylsulfone and FT206's central tetrahydronaphthalene groups below the α5 N-terminus lock the helix in nearly identical positions. Compared to the Ub-bound state, the helix is translated towards the catalytic channel and turned counterclockwise, by ~15° around the α6 N-terminus, which acts as a hinge. This raises the N-terminus of α5 by ~5 Å above the position which it would assume in the Ub-bound state and places it in a position which prevents the interaction with

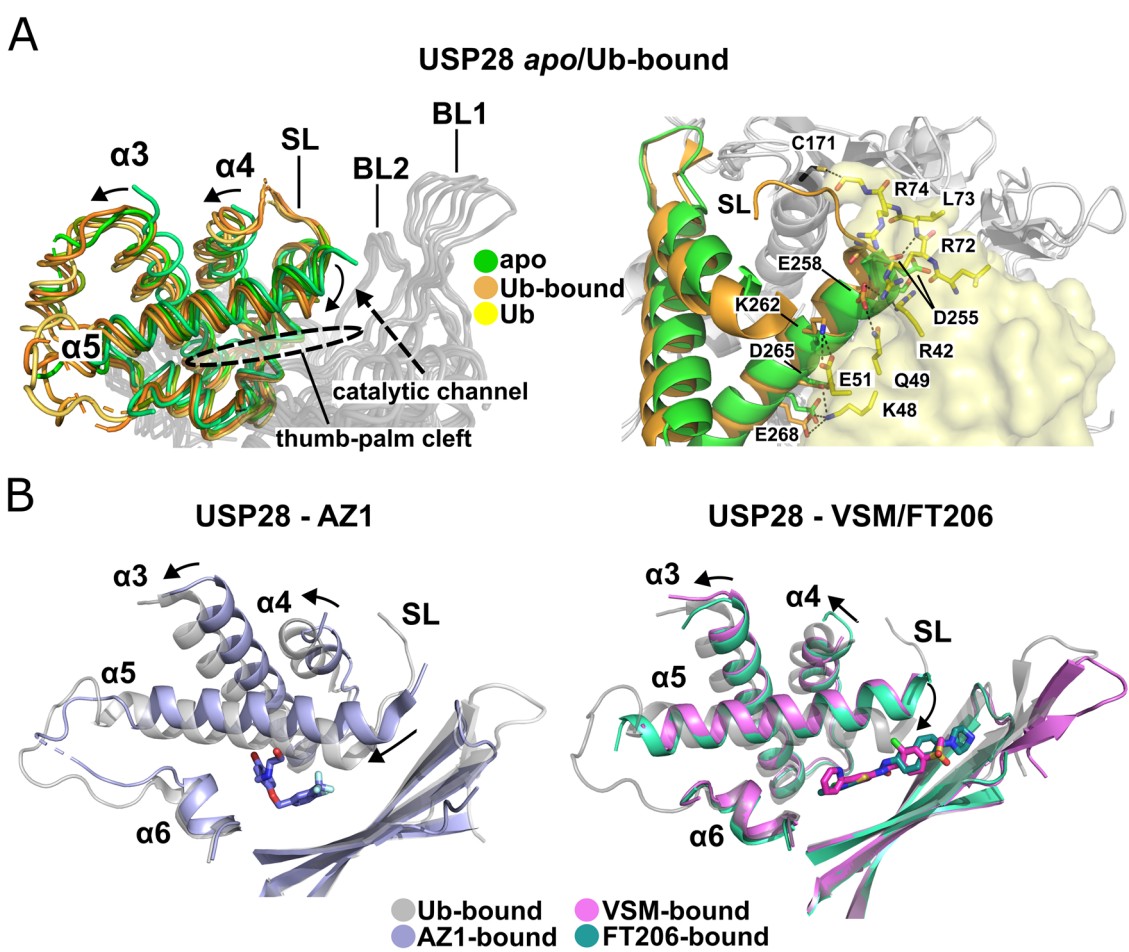

**Figure 2. Mechanism of inhibition.**

(A) Transition of the USP28 thumb from the apo- to the Ub-bound state. Left panel: Structural alignment of different USP28 apo- (green: USP28cat wt, molecules A/B, (PDB 6HEJ; Gersch et al, 2019; Data ref: Gersch and Komander, (2019b) and USP28ΔUCID, this work) and Ub-bound structures (orange: USP28cat wt–Ub, molecules A/B, PDB 6HEK and USP28ΔUCID–Ub, (PDB 6HEI) (Gersch et al, 2019; Data ref: Gersch and Komander, (2019c); Data ref: Gersch and Komander, (2019a). Structural changes in the thumb-core helices α3, α4 and α5 upon Ub-binding are indicated with arrows. Elements of the catalytic channel (switching loop: SL, blocking loops: BL1/2) are marked. Right panel: The USP28 helix α5–Ub interface. Structural alignment of USP28ΔUCID apo- (green: USP28ΔUCID) and Ub-bound structures (orange: USP28ΔUCID—Ub, PDB 6HEI) (Gersch et al, 2019; Data ref: Gersch and Komander, 2019a). Residues along helix α5 involved in hydrogen bond and salt-bridge formation with Ub are shown as sticks and hydrogen bonds as dashed lines. (B) Inhibitory mechanism: Structural alignments of USP28 Ub-bound (gray: USP28ΔUCID, PDB 6HEI) (Gersch et al, 2019; Data ref: Gersch and Komander, 2019a) and inhibitor-bound structures (blue: USP28Δtip P280H–AZ1 (left panel), pink: USP28ΔUCID wt–VSM and teal: USP28Δtip wt–FT206 (right panel)). Movement of helices α3, α4 and α5 required to reach the Ub-binding competent state from an inhibitor stabilized state are indicated with arrows. Source data are available online for this figure.

Ub Q49, R42 and R72. Furthermore, FT206 extends with its diaza-bicyclo-octane group into the catalytic channel and thereby blocks it directly.

Binding of AZ1 leads to different alterations. Compared to the substrate-bound state, helix α5 rotates by only ~5° around the α6 hinge but translates in a nearly parallel fashion by more than 2 Å away from α6 and by ~3 Å towards the catalytic channel. This more pronounced shift of the entire helix originates from an increase in the distance between the side chains of F292 on α6 and L264 on α5, that is required to form the AZ1 bromophenyl binding pocket below α5 (Fig. EV2B). This drastic displacement directly prevents Ub-tail access to the catalytic channel, by blocking the space that is required to accommodate the C-terminal Ub-residues R72, L73, and R74. Furthermore, the helix translocation affects the entire salt-bridge and hydrogen bond network that mediates Ub-binding along α5.

In addition, stabilization of α5 by the inhibitors also affects the position of the SL by indirectly altering α4 location via α3 in positions further away from the S1-site. While this is likely of minor importance in the case of AZ1, where access to the catalytic channel is directly blocked by α5, it may have more significant impact with respect to FT206 and particularly VSM where the channel is still partially accessible when the inhibitors are bound.

## Validation of the USP25-binding site

Next to USP28, AZ1, VSM, and FT206 also act as catalytic inhibitors against USP25. At present, structural data of USP25 are limited to the auto-inhibited tetrameric form, in which a large fraction of the S1-site, including the putative USP25-inhibitor-binding cleft, is occupied by the inhibitory UCID-tip (Liu et al, 2018; Sauer et al, 2019; Gersch et al, 2019). To analyze whether the

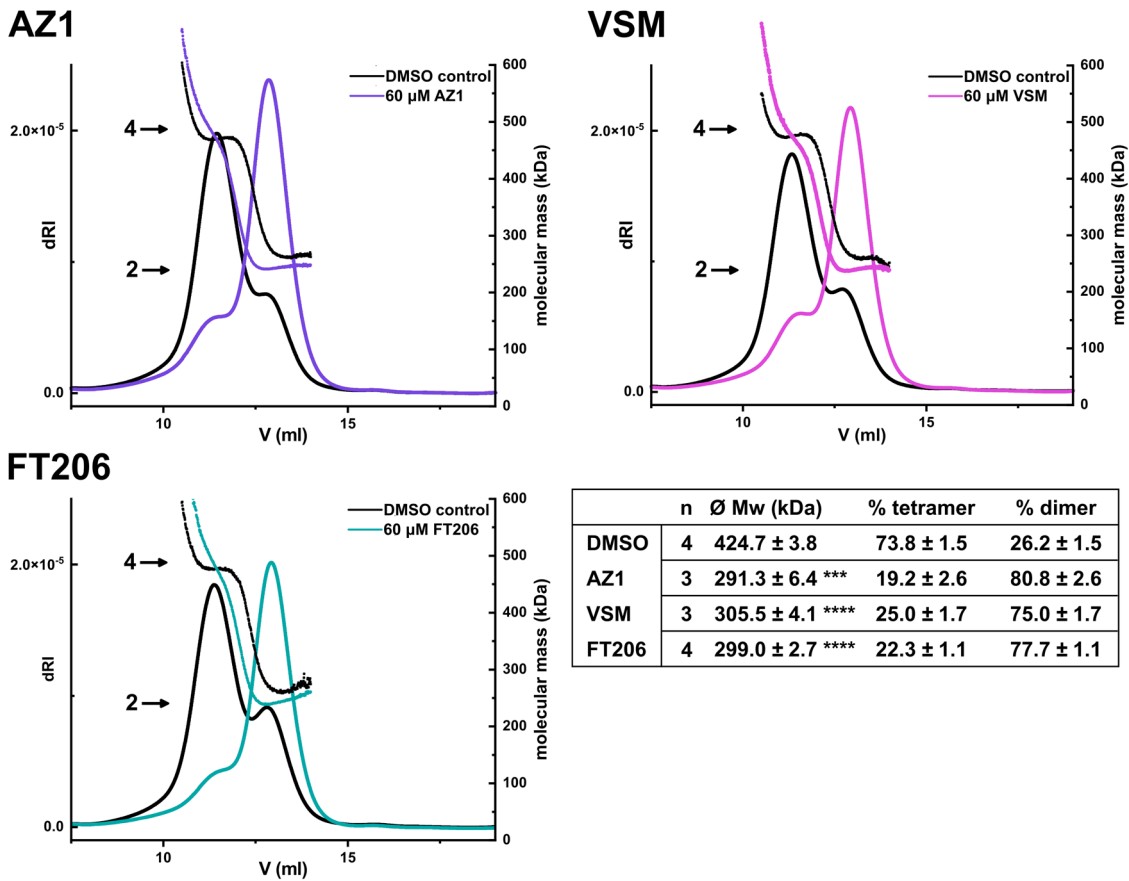

**Figure 3. Inhibitor-induced dissociation of USP25 tetramers.**

SEC-MALS analysis of 20 µM USP25fl after 45 min incubation with 60 µM AZ1 (purple), 60 µM VSM (pink) or 60 µM FT206 (teal). One representative SEC-MALS experiment is shown for each inhibitor and plotted against a control experiment (DMSO, black). Continuous lines represent the protein concentration signal (refractive index, RI). Dots show the molecular mass calculated from RI and light scattering. Arrows mark the theoretical molecular mass of tetrameric (4) or dimeric (2) USP25fl. Fractions of dimers and tetramers calculated from the average molecular masses of the entire peak fraction are given in the Table as averages of ≥3 independent experiments ± SD. Significances of differences of average molecular masses compared to the DMSO control were calculated using unpaired Student's *t* tests. Data Information: DMSO vs. AZ1: $P = 1.4*10^{-4}$; DMSO vs. VSM: $P = 4.4*10^{-6}$; DMSO vs. FT206: $P = 2.7*10^{-8}$. Source data are available online for this figure.

inhibitors also bind into the thumb-palm cleft in USP25, we investigated whether the molecules compete with the UCID-tip at the cleft in the tetrameric state. We therefore determined the oligomeric composition of full-length USP25 (USP25fl) after exposure to different concentrations of inhibitors by size exclusion chromatography coupled to multi-angle light scattering (SEC-MALS). Initial experiments, in which 10 µM USP25fl, comprised of a mixture of ~25% dimers and ~75% tetramers, were incubated with 5 µM, 10 µM or 20 µM of the three inhibitors showed that at least in the case of FT206, small-molecule induced dissociation of the tetramers into dimers occurred in a concentration-dependent manner. The fraction of dimers in the sample increased from 28% in the DMSO control to 50% and 76% after 45 min of incubation in the presence of 10 µM or 20 µM FT206, respectively. Similarly, doubling of the dimeric fraction from 23% to 46% was observed in samples containing 20 µM AZ1, while the effects of VSM were only marginal at all concentrations (Fig. EV3). At higher protein concentrations and inhibitor-to-protein ratios (60 µM inhibitor: 20 µM USP25fl dimer/tetramer mix) significant dissociation of the tetramers was reproducibly observed for all three compounds

(Fig. 3). Compared to control samples, the dimer contents increased drastically from 26.2 ± 1.5% to 80.8 ± 2.6%, 75.0 ± 1.7% or 77.7 ± 1.1% after incubation with AZ1, VSM, or FT206, respectively, thus confirming that the thumb-palm cleft represents the binding site for AZ1, VSM, and FT206 in USP25.

## Molecular basis for bi-specificity and selectivity

The disassembly of the USP25 tetramer due to the displacement of the inhibitory tip by the compounds together with the virtually identical thumb-palm cleft (Fig. 4A), provides a strong indication that the inhibitors bind to USP25 in a similar manner as observed in USP28. Small-molecule binding in a conformationally plastic thumb-palm cleft is not unique to USP28/USP25 but has previously also been shown for USP7 (Fig. EV1B) (Kategaya et al, 2017; Di Lello et al, 2017). However, the observation that the three structurally and chemically different compounds AZ1, VSM, and FT206 interact with USP28 in the same cleft, yet exert their inhibitory activity for USP28 and its close homolog USP25 without significantly affecting other DUBs of the USP family, prompted us to further investigate the origin of this selectivity.

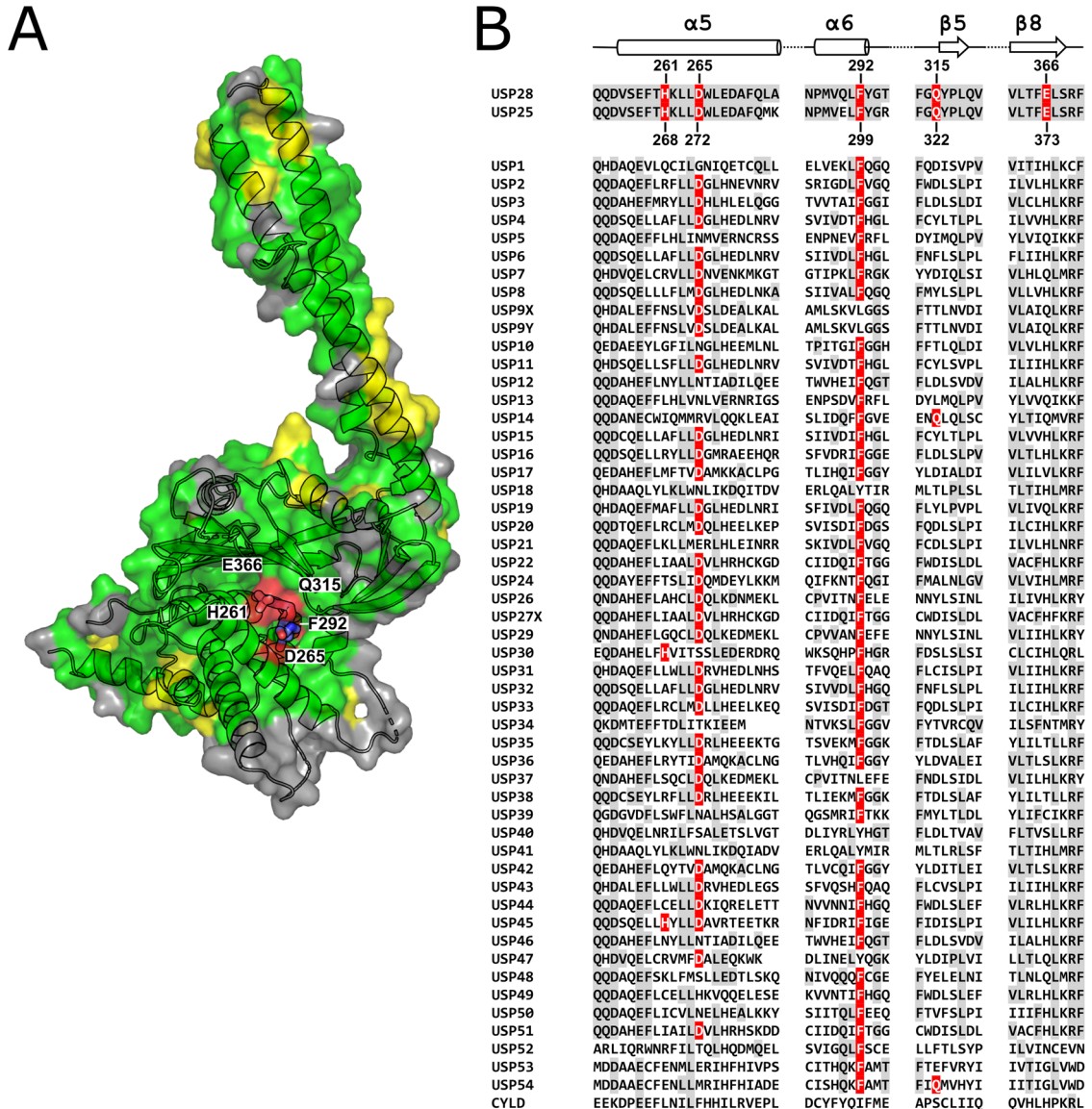

**Figure 4. Conservation of the USP inhibitor-binding region.**

(A) Sequence conservation between the USP28 and USP25 catalytic domains. Surface map of the USP28Δtip P280H–AZ1 complex structure. Residues identical in USP25 and USP28 are shown in green, type-conserved residues in yellow and non-conserved residues in gray, respectively. The AZ1 molecule in the thumb-palm cleft is shown as sticks and residues of USP28 forming hydrogen bonding or π-stacking interactions with the inhibitor as well as H261 and E366 are highlighted in red. (B) Sequence similarity between human USPs. Sequence alignment of all human USP segments representing the four secondary structures α5 (H261/H268 and D265/D272), α6 (F292/F299), β5 (Q315/Q322) and β8 (E366/E373). Residues identical to those forming bonds with the inhibitors in USP28 as well as H261/268 are highlighted in red. Other residues identical to those in USP28 are highlighted in gray. Source data are available online for this figure.

A sequence alignment of the USP28/USP25 cleft-forming region shows that out of five residues involved in direct bonding-interaction with either inhibitor, only two, D265/D272 and F292/F299, both of which interact only with AZ1, are well conserved among human USPs. Of the remaining three, a negatively charged residue equivalent to E366/E373 only exists in USP28/USP25, while glutamine or histidine, at positions corresponding to Q315/Q322 and H261/H268 respectively, each can only be found in two other USPs (Fig. 4B). We therefore hypothesized that residues E366/E373, Q315/Q322, or H261/H268 are responsible for the selectivity of the inhibitors. To test this and validate the observed or putative

inhibitor-binding sites in USP28 and USP25, we generated a series of USP28Δtip and corresponding USP25Δtip (USP25 157-464-GSGSGS-538-706) single residue variants. Thermal unfolding experiments showed that all but two variants displayed similar stabilities. Only the F292A/F299A variants showed a significant destabilizing effect (Appendix Fig. S2). Therefore, the USP28 L264F and USP25 L271F variants were generated to analyze the impact of the stacking interaction between AZ1 and F292/F299, by locking the AZ1 bromophenyl binding pocket through π-stacking. Both variants were significantly more stable than the F/A variants and therefore used for subsequent characterization. A fluorescence-

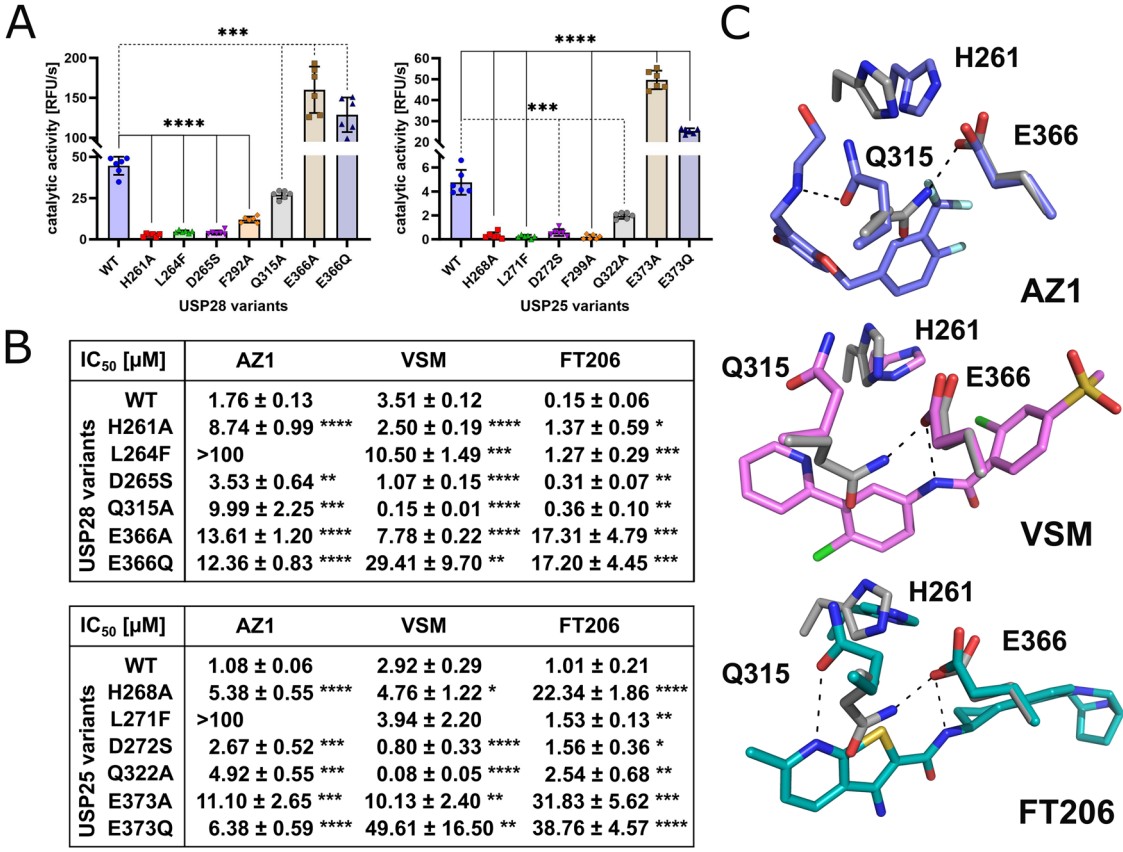

**Figure 5. Biochemical characterization of inhibition by AZ1, VSM and FT206.**

(A) Catalytic activity of USP28 and USP25 variants. Bar diagrams representing the Ub-Rh110 cleavage activity of USP28Δtip (left panel) and USP25Δtip (right panel) variants ($n \geq 5$, with two biological replicates). Corresponding variants of both DUBs are depicted in the same colors. (B) Inhibitory potencies of AZ1, VSM and FT206. The $IC_{50}$ values determined in dose–response assays with USP28Δtip and USP25Δtip variants for each inhibitor ($n \geq 5$, with two biological replicates) are summarized in the Table. (C) Displacement of the Q315 side chain upon inhibitor binding. Comparison of the positions of the three side chains (H261, Q315 and E366) by superposition of USP28ΔUCID apo (gray) and the inhibitor-bound structures (USP28Δtip P280H–AZ1 (blue), USP28ΔUCID wt–VSM (pink) and USP28Δtip wt–FT206 (teal)), from top to bottom, respectively. Hydrogen bonds between Q315 and E366 or the inhibitors are shown as dashed lines. Binding of the three inhibitors induces similar displacement of the Q315 side chain compared to its position in the apo form which requires the break of a hydrogen bond between Q315 and E366. A new bond is formed between Q315 and AZ1/FT206 but not with VSM. Data Information: Data in (A, B) are presented as average ±SD. All the statistics show the results of unpaired Student's $t$ tests, with $P$ values (*≤0.05; **≤0.01; ***≤0.001; ****≤0.0001). Source data are available online for this figure.

based mono-Ub-Rho110 hydrolysis assay was applied to determine variant activity in the absence of inhibitors. Ub-Rho110 cleavage was, likely due to modification of the USP–Ub interface or modulation of α5-mobility, moderately to strongly diminished for most variants. In contrast, a drastic increase in activity by four- and tenfold for the E366A/E373A or three- and fivefold for E366Q/E373Q substitutions was observed in USP28 and USP25, respectively (Figs. 5A and EV4A).

Next, we determined variant effects on inhibitor potency by determining $IC_{50}$ values for each variant-inhibitor-combination in dose–response assays (Figs. 5B and EV4B). AZ1 and VSM inhibited wt USP28Δtip and USP25Δtip with $IC_{50}$ values of $1.76 \pm 0.13$ μM and $1.08 \pm 0.06$ μM or $3.51 \pm 0.12$ μM and $2.92 \pm 0.29$ μM, respectively, which is in agreement with previously reported potencies (Wrigley et al, 2017; Wang et al, 2021a). FT206, for which in vitro $IC_{50}$ values have not been published, inhibited USP28Δtip with an $IC_{50}$ of $0.15 \pm 0.06$ μM and USP25Δtip with an $IC_{50}$ of $1.01 \pm 0.21$ μM, indicating a moderate selectivity towards USP28.

In agreement with the observed binding mode in USP28, mutation of residues involved in the interaction led to a reduction of AZ1 potency in both homologs. Mutation of D265/D272 and Q315/Q322, which form hydrogen bonds with the ethanolamine tail resulted in a moderate two- to six-fold increase in $IC_{50}$, indicating that this moiety plays a significant but subordinate role for AZ1 binding to both proteins. A more drastic $IC_{50}$ increase by >100-fold was observed with the L264F/L271F variants in both DUBs thus confirming that the AZ1 bromophenyl binding pocket is essential for AZ1 binding and prevention of π-stacking with F292/F299 or closure of the pocket prevents AZ1 binding.

In line with previous biochemical data (Wang et al, 2021a), mutation of Q315 in USP28 or Q322 in USP25 to alanine strongly increased the potency of VSM. Based on docking studies, this effect had been attributed to an outward flipping of the Q315 side chain that is required to accommodate VSM in the cleft. Our structures show that, in comparison to apo USP28, indeed such a flip takes place to provide sufficient space for the central chlorobenzene and

pyridine rings of VSM (Fig. 5C). Similar movements of the glutamine side chain are also required to accommodate the FT206 thienopyridine and the AZ1 fluorine-substituted ring. However, for both compounds, the Q to A mutation did not result in a strong increase but a moderate decrease in potency in either protein. A possible explanation for this can be found in the intramolecular hydrogen bond formed between the Q315 and E366 side chains in the USP28 *apo* state. This hydrogen bond needs to be broken to permit the Q315 flip. For FT206 and AZ1, the energetic costs for this are at least partially compensated by subsequent formation of a hydrogen bond with the compounds' thienopyridine respectively ethanolamine nitrogen atoms, while this is not the case for VSM, where the glutamine remains unbound (Fig. 5C).

Analogously, differences in the $IC_{50}$ values were observed upon mutation of H261/H268 and E366/E373. Removal of their side chains (variants H261A/H268A and E366A/E373A) only had a weak effect on inhibition by VSM, but led, compared to the wild type, to a moderate five-to-eleven-fold increase of the $IC_{50}$ in the case of AZ1 and a strong, nine- to more than hundred-fold increase for FT206. This suggests that the hydrogen bond between FT206 and E366/E373, respectively, the multipolar interactions between H261/H268 and AZ1 or cleft locking H261/H268–E366/E373 arc, play an important role for binding and inhibition of both molecules. In contrast, release of the steric block between E366/E373 and Q315/Q322 may compensate for the loss of the hydrogen bond between E366/E373 and VSM. To further decipher the role of E366/E373, we generated the additional variants E366Q/E373Q which lack the negative charge of the glutamate side chain, while retaining the ability to bind Q315/Q322 through hydrogen bonds. These variants displayed strongly increased catalytic activity and impairment of AZ1 and FT206 inhibition comparable to their alanine counterparts. However, both glutamine variants also suppressed inhibition by VSM to a similar extent as seen for the other inhibitors, thus emphasizing the importance of the negative charge provided by the E366/E373 carboxyl group in this position for inhibition.

Taken together, mutation of the observed USP28-inhibitor interface and its putative equivalent in USP25 led to modulation of all three inhibitor potencies, which confirms their binding in the thumb-palm cleft of both enzymes. Mutation of the two poorly conserved residues, Q315/Q322 and H261/H268, but more importantly the unique E366/E373 had a significant impact on inhibition, thus emphasizing their important role in the compounds' selectivity.

### Role of glutamate 366/373

The increase in catalytic activity by the E366A/E373A mutation and its central role for inhibition demanded further investigation. For this purpose, MD simulations of mutated and wild-type-like USP25ΔUCID and USP28ΔUCID models were performed. As there is no crystal structure of USP25 in an activated state, a homology model with a composition corresponding to USP28ΔU-CID was prepared based on the auto-inhibited, tetrameric USP25 (PDB 6H4J) (Sauer et al, 2019; Data ref: Klemm et al, 2019a). For each system, six independent runs with 250 ns were carried out, resulting in a total sampling time of 1.5 μs for each protein.

In the wild-type-like simulations of USP28 and USP25, E366/E373 form several interactions with side chains from α5, either directly or mediated by solvent water or a sodium ion (Fig. 6A).

Key interaction partners are S257 and E258 or S264 and E265 in USP28 and USP25, respectively. These contacts lead to the two observed maxima in the distribution at approximately 9 Å and 10.5 Å as observed in the histogram shown in Fig. 6B displaying the distance between the Cα of E366/E373 (A) and the N-terminal turn of α5 for USP28 and USP25, respectively. However, without these interactions acting as a placeholder, the helix can shift significantly closer to the adjacent β-strands, resulting in a new maximum in the distribution at about 7 Å. Slight differences in the observed distance distributions between USP25 and USP28 in both, mutated and unmutated forms can be attributed to the different composition and length of the switching loop adjacent to α5. For example, the stacking interaction between F253 and F266 present in USP25 is missing in USP28, while in USP28 E251 provides an additional negative charge close to the α5 N-terminus.

To illustrate the distinct positions of the helix, a clustering focusing on the ten N-terminal amino acids of α5 was performed. For the wild-type constructs, the main clusters of USP28 and USP25 are representative for 59% and 72% of the frames, respectively, and are close to the apo structure. The centroid frames of each cluster are shown here as snapshots (Fig. EV5A). The USP28 snapshot captures the hydrogen bond network between H261 and Q315, as known from crystal structures, while the USP25 snapshot shows exemplarily a bridging sodium ion between S264, E265, and E373 (Fig. 6B).

Although the main cluster for the set of E366A/E373A mutant simulations is similar to the clusters described above, the second most populated clusters (cluster 2) of the mutated proteins represent a shifted conformation, as expected from the distance histograms (Figs. 6B and EV5B). This new conformation is additionally confirmed by a principal component analysis (PCA) (Fig. EV5C). Thus, the E366A/E373A mutation enables a repositioning of helix α5 without the loss of hydrogen bonds or ionic interactions prevalent in the wild-type forms between α5 and the adjacent β-strands. We therefore suggest that the increased catalytic activity in the variant is related to this facilitated shift of helix α5, which possibly supports the binding of ubiquitin.

## Discussion

The decisive role of USP28 and USP25 in maintaining elevated cellular levels of different 'hard to drug' cancer-promoting proteins has rendered both DUBs highly attractive targets for the development of small-molecule inhibitors against different cancers (Popov et al, 2007; Wu et al, 2013; Diefenbacher et al, 2014; Xu et al, 2017; Prieto-Garcia et al, 2020; Nelson et al, 2022). Several potent small-molecule inhibitors targeting USP28 have been identified and where tested, they have been found to act against USP28 and USP25 activity with similar potency, while being selective over other DUBs (Wrigley et al, 2017; Liu et al, 2020; Wang et al, 2021a; Ruiz et al, 2021; Varca et al, 2021; Xu et al, 2023). This finding may potentially limit further pharmacological development of these bispecific molecules due to side effects linked to USP25's regulatory role in different metabolic and immune system-related processes (Zhong et al, 2013; Lin et al, 2015; Habtemichael et al, 2018; Wen et al, 2019; Liu et al, 2022).

Here we shed light on the origin of the bi-specificity and selectivity observed for many of the known inhibitors by

characterizing the mode of inhibition of both DUBs utilizing the established small-molecule inhibitors AZ1, VSM and FT206.

The structures of the USP28-inhibitor complexes revealed that the three molecules occupy different positions in the same, extensive hydrophobic cleft, located between the thumb and palm subdomains, within the S1-site that acts as binding surface for the globular Ub-domain and harbors the auto-inhibitory tip in USP25. (Fig. 1A). The cleft-forming residues in USP28 and USP25 are virtually identical in sequence (Fig. 4A,B), thus providing a direct explanation why both DUBs are inhibited in a similar manner.

All three inhibitors act by a common allosteric mechanism. They block the required conformational transition of the USP28/ USP25 thumb by blocking the movement of α5 and thus the reshaping of the USP S1-site into a substrate-binding competent state (Fig. 2A,B). This mechanism is in agreement with HDX-MS results by Wang et al, where next to the peptide sequences coinciding with the observed inhibitor-binding site, lower, but significant changes in hydrogen deuterium exchange rates have also been measured in helices α3 and α4 which are not part of the cleft. The structural dynamic of the thumb-helices is likely the reason why previous predictions of the ligand pose based on molecular docking of AZ1 and VSM significantly deviate from those observed in our crystal structures, as docking methods cannot model the

shift of α5 observed upon ligand binding (Liu et al, 2020; Wang et al, 2021a). Our results therefore provide the structural basis for the future design of USP28 inhibitors.

According to the DUB-inhibitor classification system recently proposed by Lange and colleagues, the USP28/USP25 inhibitors belong to the most frequently found class III-D, which consists of molecules binding to the S1-site and act by blocking the distal Ub from binding to the DUB (Lange et al, 2022). Most of the known USP-inhibitors of this class and class II* bind in the structurally highly dynamic catalytic channel either non-covalently (class III-D) or covalently (class II*) (Báez-Santos et al, 2014; Turnbull et al, 2017; Lamberto et al, 2017; Gavory et al, 2018; Wang et al, 2018). Only few molecules, such as the inhibitors GNE6640/6776 and Cpd2 are known to engage the thumb-palm cleft (Kategaya et al, 2017; Di Lello et al, 2017). All three lock the USP7 helix, corresponding to USP28 α5 in a position nearly identical to that observed in its apo state and thereby likely prevent its movement towards the catalytic channel which is required, among other rearrangements, to establish the Ub-binding competent S1-site. However, in USP7 the same blockage has also been achieved by molecules binding into the catalytic channel such as FT827 or ALM2 (Fig. EV1B) (Gavory et al, 2018; Turnbull et al, 2017).

Although our investigation is limited to three compounds, the finding that all bind to USP28/USP25 in the thumb-palm cleft

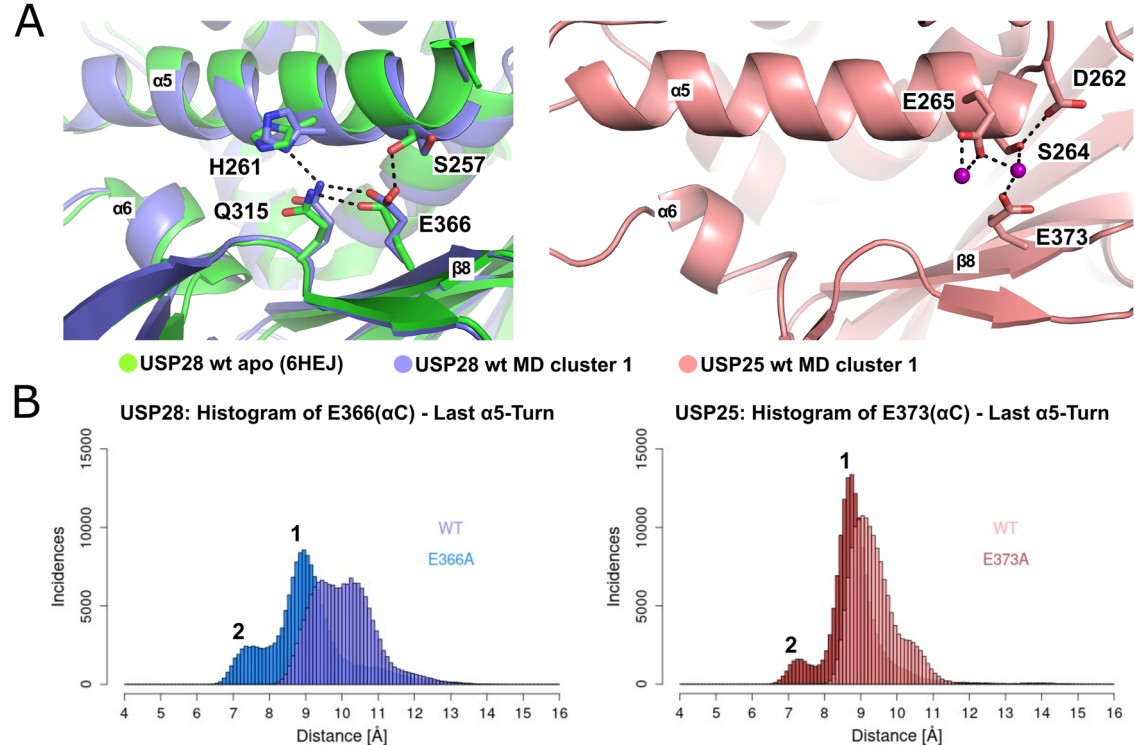

**Figure 6. Role of glutamate 366/373 for α5-mobility.**

(A) Closeup view on helices α5. The hydrogen bonding network of E366/E373 of the representative snapshots from cluster 1 of the MD simulations of USP28 and USP25 is shown. The left panel displays superpositions of USP28ΔUCID (green) and the representative snapshot of cluster 1 from USP28ΔUCID (blue) while the right displays the snapshot of USP25ΔUCID (pink). The hydrogen bond network of E366/E373 is highlighted. Corresponding residues are shown as sticks, hydrogen bonds as black dashed lines and sodium ions as purple spheres. Secondary structure elements containing residues mediating bonds with inhibitor (α5, α6, and β8) are marked. (B) E366/E373 influences the mobility of α5. Histograms representing the distances between the Cα of E366/E373 and the geometrical center of the proximal N-terminal turn of helix α5 (residues V256/V263 to F259/F266). The distance measurements are based on the MD simulations conducted with wt (light-) and E366A/E373A (dark-colored) ΔUCID variants of USP28 (left) and USP25 (right panel), respectively. The histogram shows maxima at ~7.5 Å and ~9 Å, which were clustered as clusters 1 and 2, respectively, for consecutive analysis. Source data are available online for this figure.

suggests, that it acts as a 'molecular sink' which sets both enzymes apart from other USPs for which inhibitors have previously been identified. Its size and structural plasticity provides a conceivable explanation for why USP28 has been found to be highly susceptible for small-molecule inhibition when compared to other proteins of the family (Varca et al, 2021).

Sequence comparison and biochemical analysis of enzyme mutant variants allowed us to trace the molecular origin of this susceptibility. Out of five residues involved in bond formation with any of the inhibitors or locking the cleft towards the protein surface through interaction with each other, three (H261/H268, Q315/Q322 and E366/E373) are poorly conserved within the USP family (Fig. 4B).

Mutational analysis of these residues confirmed the observed, respectively assumed binding sites for each inhibitor in both proteins and reflected differences in the individual protein-inhibitor interfaces such as the previously reported steric interference between Q315/Q322 and VSM (Fig. 5C) (Wang et al, 2021a). Our analysis also revealed that E366 (USP28) and E373 (USP25) which provide a unique negative charge at the catalytic channel proximal end of the cleft, play a role for inhibition that lies beyond that of a direct and indirect interaction partner for the inhibitors as anticipated from the structures. In agreement with our structural observations, mutation of this position to alanine led to weak (VSM), moderate (AZ1) or strong (FT206) reduction of inhibitor potencies for both DUBs. However, similar (AZ1, FT206) or even stronger (VSM) reduction was observed upon E366/E373 mutation to glutamine where the glutamate side chain–inhibitor and side chain–side chain interactions would be expected to be at least partially preserved (Figs. 1B and 5C). Further analysis pointed towards an additional role of glutamate as a global 'structural facilitator' of inhibition. The drastic increase in cleavage activity of the E366/E373 variants compared to the wt proteins (Fig. 5A) and the MD simulations of the wt and E366A/E373A variant proteins (Fig. 6B) indicate that the carboxyl group of E366/E373 efficiently stabilizes an "outward" position of helix α5 by mediating different direct and indirect interactions with this helix (Fig. 6A). This stabilization suppresses the transition of the S1-site from an auto-inhibited "cleft open", to a catalytically active "cleft closed" state suitable for Ub-substrate binding. The glutamate's cleft stabilization may further explain why FT206 is more strongly affected than AZ1 with respect to the E366/E373 variants. FT206 being the more extended and rigid molecule which forms only two hydrogen bonds with the DUB may require a more stable, open state to be accommodated in the cleft. The smaller AZ1, in contrast, forms more bonds with the DUB and can be more readily positioned due to its compactness and flexibility.

These observations raise the question what the functional role of the suppression of cleft closure by E366/E373 is. USP25 exists as a stable, auto-inhibited tetramer (Liu et al, 2018; Sauer et al, 2019; Gersch et al, 2019). While we show that all three inhibitors can mechanistically compete with the cleft-bound UCID-tip resulting in dimeric USP25, our experimental setup does not allow conclusions that such dissociation would occur at lower inhibitory concentrations. However, Liu et al identified E373 as an essential residue for tetramer formation (Liu et al, 2018). Our data link this finding and inhibition, as E373 promotes a cleft-open state which facilitates the binding of the UCID-tip and the here investigated compounds. Given that USP28 only exists as an active dimer, one may speculate that USP28 and USP25 have evolved from a common tetrameric ancestor USP. USP28 then subsequently lost the ability to tetramerize, for example by mutation of the UCID-tip, while the auto-inhibitory residue E366 has been preserved as a means of activity regulation.

Our analysis has important implications for the future development of USP28 and USP25 inhibitors. While moderate selectivity for one DUB over the other appears to be achievable, as evidenced by the slightly higher potency of FT206 for USP28 compared to USP25, the high similarity of the binding cleft between both enzymes suggests that achieving "absolute" selectivity of a cleft-binding inhibitor for either DUB will be a challenge. Structural data of USP25 in an active and inhibitor-bound state to delineate structural differences between the DUBs, which are currently not available, would be highly desirable to support such development.

The strong dependence of inhibition by the three molecules and most likely other cleft-binding inhibitors on a single residue E366 (USP28) or E373 (USP25) creates a 'one defeats all' situation where any molecule binding to this site will likely be rendered ineffective by mutation of the glutamate. This problem is potentiated by the mutational burden cells in many cancers are exposed to (Alexandrov et al, 2013). Furthermore, the issue becomes more severe, given that the cancer-promoting activity of USP28 and USP25 will probably be further stimulated by the catalytic hyperactivity arising from mutation of the respective glutamates. This suggests that mutated cells will possess a double selective advantage compared to cells carrying non-mutated USP28 or USP25.

The issues arising from inhibition by cleft binding and the high similarity of both DUB's S1-sites can possibly be circumvented by a search for new inhibitors towards alternative binding sites. The here described USP28 and USP25 variants can provide a useful filtering tool to accelerate such a focused search within compound libraries used for de novo screening or collections of existing USP28/USP25 inhibitors.

## Methods

### Inhibitors

Vismodegib was obtained from Hycultec (Beutelsbach GER; cat. # HY-10440). AZ1 was provided by AstraZeneca (Cambridge UK). Racemic FT206 was synthesized as described below. All inhibitors were stored as powders and dissolved in 100% DMSO at 10 mM stock concentration prior to first usage. Stock solutions were stored at $-20\,°C$.

### Synthesis of racemic FT206

To record NMR-spectra, compounds were dissolved in $CDCl_3$ or DMSO-$d_6$ and measured on Avance 300 or Advance 400 from Bruker Corporation (Massachusetts, USA). All chemical shift values are reported in ppm, the multiplicity of the signals is assigned as follows: s (singlet), d (duplet), t (triplet) and m (multiplet). Mass spectrometry analysis was performed in positive ion mode by electrospray-ionization (ESI) on a LCMS-2020 single quadrupole MS from Shimadzu (Duisburg, Germany). Precision mass was measured using a MALDI Orbitrap XL from Life

Technologies GmbH (Darmstadt, Germany). For purity estimation of the synthesized compounds, a reverse-phase high-performance liquid chromatography (RP-HPLC) was performed using the Luna 10 µm C18(2) 100 Å, LC Column 250 × 4.6 mm from Phenomenex LTD (Aschaffenburg, Germany) and the analysis was conducted using the Shimadzu prominence module. Acetonitrile and aqueous formic acid 0.1% were used as eluents. The established method for purity determination was initiated with 90% water (0.1% formic acid), then a linear gradient from 90% to 5% water (0.1% formic acid) for 13 min was chosen, and finally additional 7 min 5% water (0.1% formic acid). The flow rate was adjusted to 1.0 mL/min and the UV–vis detection was performed at 254 nm and 280 nm, respectively.

a) $Pd(OAc)_2$, XPhos, $Cs_2CO_3$, Toluene, 105 °C, quantitative b) HCl 4 M in dioxane, quantitative c) HBTU, EDCl, 4-methylmorpholine, DMF, 45% d) $BBr_3$, DCM 47%.

**Step a:** *tert*-Butyl (6-bromo-1,2,3,4-tetrahydronaphthalen-2-yl) carbamate (108 mg, 0.33 mmol) was dissolved in anhydrous toluene (5.0 mL) under Ar. To the resulting solution, benzyl 3,8-diazabicyclo[3.2.1]octane-8-carboxylate (98.5 mg, 0.40 mmol), palladium acetate (7.56 mg, 0.033 mmol), XPhos (32.4 mg, 0.066 mmol) and cesium carbonate (332 mg, 1.0 mmol) were added. The reaction mixture was stirred overnight at 105 °C and then cooled to RT and quenched with water. The resulting mixture was extracted with ethyl acetate 3x. The combined organic layers were washed with brine, dried over anhydrous magnesium sulfate, filtered, and concentrated under a vacuum. The resulting crude product was purified by flash chromatography (hexane/EtOAc, 9:1 to 7:3) to yield 162 mg (quantitative) of **1**. ESI: 492.5 $[M + H]^+$.

**Step b:** Compound **1** (162 mg, 0.33 mmol) was dissolved in 5 mL 4 M HCl in dioxane and the resulting solution was stirred for 2 h at RT. The reaction mixture was concentrated under vacuum to afford compound **2** which was used in the next step without further purification (143 mg, quantitative). ESI: 392.8 $[M + H]^+$.

**Step c:** 3-Amino-6-methylthieno[2,3-b]pyridine-2-carboxylic acid (62.5 mg, 0.30 mmol) and HBTU (172 mg, 0.45 mmol) were dissolved in 4.0 mL DMF under Ar and stirred for 5 min at RT. Then the solution was cooled to 0 °C and **2** (143 mg, 0.33 mmol) in 2.0 mL DMF and 4-methylmorpholine (198 µL, 1.80 mmol) were added. The resulting solution was stirred at 0 °C for 30 min. Finally, EDCl (86.3 mg, 0.45 mmol) was added at 0 °C and the reaction mixture stirred at RT overnight. DMF was evaporated and the residue was suspended in a saturated aqueous sodium bicarbonate solution and ethyl acetate. The phases were separated, and the water layer extracted again with ethyl acetate (2×). The combined organic layers were dried over magnesium sulfate, filtered and evaporated. Purification by flash chromatography (hexane/EtOAc, 7:3 to 0:1) to yield 78.0 mg (45%) of **3**. $^1$H NMR (400 MHz, DMSO-$d_6$) δ 13.71 (br s, 1H), 8.30 (d, $J$ = 8.3 Hz, 1H), 7.53 (d, $J$ = 8.3 Hz,

1H), 7.40–7.32 (m, 5H), 7.14 (br s, 2H), 6.90 (d, $J$ = 8.5 Hz, 1H), 6.65 (dd, $J$ = 8.7, 2.2 Hz, 1H), 6.57 (d, $J$ = 2.2 Hz, 1H), 5.11 (s, 2H), 4.33 (br s, 2H), 4.15–4.05 (m, 1H), 3.47–3.42 (m, 2H), 3.17 (d, $J$ = 2.5 Hz, 1H), 2.87–2.66 (m, 5H), 2.58 (s, 3H), 1.99–1.68 (m, 6H); ESI: 583.3 $[M + H]^+$.

**Step d:** Boron tribromide 1 M in DCM (2.0 mL, 2.0 mmol) was added dropwise to a round bottom flask containing **3** (70.0 mg, 0.120 mmol) in 1.0 mL DCM at −20 °C under Ar. The resulting solution was allowed to reach RT and was stirred for 1 h. MeOH (3.0 mL) was added to quench the reaction. The solvent was evaporated, and the residue was purified by preparative HPLC (water with 0.1% formic acid and acetonitrile) to yield 25.1 mg (47%) of compound FT206 as formic acid salt. $^1$H NMR (300 MHz, DMSO-$d_6$) δ 8.37 (br s, 1H), 8.31 (d, $J$ = 8.2 Hz, 1H), 7.54 (d, $J$ = 7.7 Hz, 1H), 7.30 (d, $J$ = 8.3 Hz, 1H), 7.16 (br s, 2H), 6.91 (d, $J$ = 8.4 Hz, 1H), 6.63 (d, $J$ = 8.5 Hz, 1H), 6.56 (s, 1H), 4.17–4.04 (m, 1H), 3.81 (br s, 2H), 3.43 (d, $J$ = 11.7 Hz, 2H), 2.88–2.68 (m, 6H), 2.58 (s, 3H), 1.98–1.71 (m, 6H); $^{13}$C NMR (75 MHz, DMSO-$d_6$) δ 165.4, 164.6, 159.0, 158.1, 148.7, 145.5, 135.7, 130.7, 129.3, 125.2, 124.0, 119.3, 113.8, 112.4, 96.4, 53.4, 52.3, 45.8, 34.3, 29.3, 28.9, 26.9, 24.2; $R_f$ HPLC: 7.3 min (13 min from 10 to 95% MeCN in water (0.1% formic acid), then 7 min 95% MeCN). 98.9% purity; HRMS (MALDI): m/z found. 448.2162 $[M + H]^+$ (cal. $C_{25}H_{30}N_5OS$ 448.2166).

## Cloning

Constructs of human USP28 and USP25 used in this study were amplified by PCR using S7 Phusion polymerase (MOBIDIAG) from constructs of the full catalytic domains and cloned between the 3C-cleavage and BamH1 restriction site of the vectors pCDFM-14 or pCDFM22 (in house vectors carrying T7 expression cassettes identical to pETM-14 and pETM-22 (EMBL-plasmid-collection)) with an N-terminal 6xHis, or a thioredoxin-6xHis-tag, respectively, using the SLIC method (Li and Elledge, 2007). Point mutations were introduced using the QuickChange site-directed mutagenesis kit (Stratagene) according to the manufacturer's instructions. The USP28ΔUCID construct was generated by linearization and re-circulation by PCR, thereby replacing the full UCID (amino acids 400–580) with a triple GS linker. The USP28Δtip wt and USP25Δtip wt constructs have previously been described (Sauer et al, 2019).

## Recombinant protein expression and purification

All USP28 and USP25 proteins were expressed in *E. coli* BL21star (DE3) cells (Invitrogen) carrying a pRARE2 rare codon plasmid (Novagen). Cells were grown in TB medium supplemented with 50 µg/mL streptomycin and 34 µg/mL chloramphenicol at 37 °C and 200 rpm to an $OD_{600}$ of 2.0–3.0. Expression was induced by the addition of IPTG to a final concentration of 0.3 mM, and cells were further shaken at 20 °C for 16–20 h. Cells were harvested by centrifugation at 4000 × $g$ and 4 °C for 20 min. Pellets were resuspended in 10 mL/g lysis buffer (50 mM HEPES pH 8.0, 300 mM NaCl, 1 mM TCEP) supplemented with DNase I (1 U/mL), lysozyme (0.5 mg/mL) and one cOmplete™ EDTA-free protease inhibitor cocktail tablet (Roche) per 200 mL volume. Cells were lysed using a LM20 cell disruption system (Microfluidizer) at 1.5 kbar, and the crude extract was cleared by centrifugation at

30,000 × g and 4 °C for 60 min. The lysate was filtered through a 0.45-µm syringe filter and loaded onto a column containing Protino® Ni-IDA beads pre-equilibrated with the lysis buffer. The column was washed with 10 column volumes (CV) of high salt wash buffer (50 mM HEPES pH 8.0, 1 M NaCl, 1 mM TCEP), and the protein was eluted with lysis buffer supplemented with 400 mM Imidazole (pH 8.0). 3C-Protease was added in a 1:50 ratio to cleave the tag, and the protein was dialyzed overnight at 4 °C against lysis buffer.

The dialysate was concentrated using an Amicon® centrifugal device and further purified by size exclusion chromatography (SEC) (HiLoad™ 16/600 Superdex™ 75 pg (Cytiva)), pre-equilibrated with SEC buffer (20 mM HEPES pH 8.0, 150 mM NaCl, 1 mM TCEP) utilizing an ÄKTA™ pure system (GE Healthcare), equipped with UNICORN™ 7.7 software for data analysis (Cytiva).

The USP25fl protein was purified similarly, but dialysis was performed against a lower salt concentration (150 mM NaCl). Subsequently, the proteins were threefold diluted with buffer A (50 mM HEPES pH 8.0, 1 mM TCEP), loaded onto an anion exchange chromatography column (MonoQ 10-100-GL (Cytiva)), and eluted with a linear gradient from 50 to 500 mM NaCl over 22.5 CV with buffer B (50 mM HEPES pH 8.0, 1 M NaCl, 1 mM TCEP). Separation of the oligomeric species of USP25fl and final purification were carried out by size exclusion chromatography using a Superose 6 XK 16–70 pg column (Cytiva), pre-equilibrated with SEC buffer. The elution fractions were analyzed by SDS-PAGE, and fractions containing the pure protein were concentrated and flash-frozen in liquid nitrogen prior to storage at −80 °C.

## Protein crystallization and inhibitor soaking

USP28Δtip crystals were grown in a hanging drop vapor diffusion setup by mixing 2 µL of purified protein (2.5 mg/mL) with 2 µL of reservoir solution (0.5 M sodium malonate pH 6.0, 0.1 M sodium citrate). Crystals appeared within 2–5 days at 20 °C and reached their maximum size (ca. 200 µm × 200 µm × 100 µm) within one additional week.

Crystals of USP28ΔUCID were grown in a sitting drop vapor diffusion setup by mixing 300 nL protein (15 mg/mL) with 150 nL reservoir solution (0.05 M NaCl, 0.1 M Li₂SO₄, 0.1 M MES pH 6.4, 14% w/v PEG 10,000). Crystals appeared within 1–2 days and grew to their maximum size within a further 2 days.

For inhibitor soaking, the pre-formed USP28Δtip or USP28ΔUCID crystals were taken out of their original crystallization drop and placed for 3 h into a 1 µL drop of the respective inhibitor diluted from a 10 mM stock (in 100% DMSO) in the reservoir solution reaching a final inhibitor concentration of 150 µM. The final DMSO concentration in the inhibitor drop was 1.5%. After soaking, the crystals were transferred into a cryoprotectant solution consisting of the respective inhibitor and mother liquor supplemented with 25% (v/v) glycerol (USP28Δtip) or 22% (v/v) PEG400 (USP28ΔUCID) and then flash-frozen and stored in a loop in liquid nitrogen until data collection.

## Data collection, structure determination, and refinement

Diffraction data for USP28Δtip crystals soaked with AZ1 or with FT206 were collected at BESSY Beamline 14.1 and EMBL/DESY Beamline P14, respectively. Diffraction data for USP28ΔUCID apo

or in complex with VSM were collected at ESRF Beamline ID23-2. Data integration and scaling were performed with XDS (Kabsch, 2010) and Aimless (USP28ΔUCID data) (Evans and Murshudov, 2013) or Staraniso (USP28Δtip data) (Tickle et al, 2018), respectively. The structures were solved by molecular replacement with Phaser using chain A of the USP28cat apo structure (PDB 6H4I) (Sauer et al, 2019; Data ref: Klemm et al, 2019b)) as the search model. The models were then manually rebuilt with Coot (Emsley et al, 2010). Automated refinement of model coordinates, TLS parameters and B-factors was carried out with PHENIX-refine (Adams et al, 2010) using inhibitor coordinate restraints generated with Grade (Smart et al, 2021). Data collection, phasing and refinement statistics are listed in Table 1.

## Structural analysis and visualization

Protein-inhibitor and protein-protein interfaces were analyzed using Coot (Emsley et al, 2010), LigPlot+ (version 2.2) (Laskowski and Swindells, 2011) or PISA (Krissinel and Henrick, 2007), respectively. Electron density maps were generated using the phenix.refine module (Adams et al, 2010). Simulated annealing (SA) composite omit maps were calculated using the Phenix Composite Omit map tool by running 20 simulated annealing cycles with the refined USP28-inhibitor models in each of which 5% of the protein atoms and the inhibitor molecules had been omitted. Structural figures were generated with Pymol (Schroedinger LLC, open source version 2.6).

## SEC-MALS analysis

Multi-angle light scattering (MALS) analysis combined with SEC was used to examine the effects of inhibitor binding on the oligomeric state of USP25fl. Two sets of experiments were carried out. For the first set, 75 µL of 20 µM USP25fl (USP25 1-1055) in size exclusion buffer (20 mM HEPES pH 8.0, 150 mM NaCl, 1 mM TCEP) were mixed with 75 µL of 10 µM, 20 µM or 40 µM AZ1, VSM or racemic FT206 dilutions in size exclusion buffer containing 0.4% (v/v) DMSO or size exclusion buffer with 0.4% (v/v) DMSO as a control. For the second set, 75 µL of 40 µM USP25fl in size exclusion buffer were mixed with 75 µL of 120 µM AZ1, VSM or racemic FT206 dilutions in size exclusion buffer containing 1.2% (v/v) DMSO or size exclusion buffer with 1.2% (v/v) DMSO as a control. Final resulting inhibitor concentrations were 5 µM, 10 µM and 20 µM (first set) or 60 µM (second set). Samples were incubated at 4 °C for 15 min and then centrifuged for 30 min at 4 °C at 30,000 × g. 100 µL of the samples were then injected onto a Superose 6 10/300 GL column (Cytiva) equilibrated with size exclusion buffer at a constant flow rate of 0.5 mL/min. Light scattering and concentration signals of the eluates were detected using a Dawn Heleos 8+ light scattering detector and an Optilab T-rEX refractive index detector (Wyatt Technologies). Data analysis was carried out using the ASTRA software (version 6.1.5.22, Wyatt Technologies) and the data were plotted with OriginPro (version 2021b, OriginLab). Ratios of tetrameric and dimeric USP25fl species were calculated using the following equations:

$$n(tetramer) = \frac{experimental\ MW}{theoretical\ MW(USP25fl\ dimer)} - 1$$
$$1 - n(tetramer) = n(dimer)$$

## Differential scanning fluorimetry

Protein stability of the USP28 and USP25 Δtip variants was analyzed by differential scanning fluorimetry to determine their melting temperature ($T_m$). In total, 1 μL of the wt or the purified variants (5 mg/mL) were supplemented with 25x SYPRO Orange dye in 25 μL SEC buffer in a 96-well PCR plate. Thermal unfolding was analyzed with a real-time PCR cycler (Stratagene Mx3005P, Agilent Technologies) from 25 °C to 95 °C with an increment of 1 °C/min. Fluorescence was measured at excitation and emission wavelengths of 490 nm and 575 nm, respectively. Data evaluation and determination of the melting temperature were performed using the MxPro qPCR software (Agilent Technologies).

## Dose–response assays

The cleavage activity of USP28 and the USP25 Δtip variants were measured in the presence and absence of inhibitors utilizing the fluorogenic substrate, Ub-Rhodamine110Gly (Ub-Rho110, UbiQ Bio). For $IC_{50}$ (half maximal inhibitory concentration) determinations, AZ1, VSM, and FT206 were diluted from a 10 mM stock (in 100% DMSO) with the assay buffer (20 mM HEPES pH 7.5, 150 mM NaCl, 1 mM TCEP and 50 μg/mL BSA) in a threefold, ten-point dilution series, starting from 100 μM. The variants (final concentration: 20 nM) were incubated with the respective compound for 15 min at room temperature. Due to large differences in variant activities compared to the wt proteins, the protein concentrations in the dose–response assay were adjusted for some variants (final concentration: 100 nM for variants H261A/H268A and 10 nM for both, E366A/E373A and E366Q/E373Q). The components of the assay were pipetted in a black non-binding 384-well plate (Greiner Bio-One) and the cleavage reaction was initiated by the addition of Ub-Rho110 (final concentration: 250 nM) to the DUB-inhibitor mixture. Fluorescence was measured using the CLARIOstar microplate reader (BMG Labtech) at 25 °C (excitation and emission wavelength at 485 nm and 585 nm, respectively) for 30 min. The initial slope of the increasing fluorescence signal was measured, and the relative inhibition was determined and plotted by non-linear regression against the inhibitor concentration utilizing the GraphPad Prism (version 9.1.2) software. All statistical results are presented as the mean ± SD. $P$ values (*≤0.05; **≤0.01; ***≤0.001; ****≤0.0001) were obtained using unpaired Student's $t$ test, based on 2–3 technical replicates of two batches from independently purified proteins ($n \geq 5$).

## MD simulations

Two crystal structures were used as starting points for the MD simulations: USP25 (PDB 6H4J) (Sauer et al, 2019; Data ref: Klemm et al, 2019a) and a monomeric USP28ΔUCID structure (this work, 8P19). As there is currently no crystal structure of monomeric USP25ΔUCID, a homology model was derived with MOE based on the tetrameric USP25 structure (Molecular Operating Environment (MOE), 2022.2, 2022). For this approach, residues from R407 to M586 were deleted and a GSGSGS-motif was inserted instead. 100 possible backbone conformations were generated de novo for the new loop H406-GSGSGS-I587. For the top five models, side chains

were added. Then, the energetically best scored model was chosen from this selection. For other, smaller loops or side chains that were missing in the USP25ΔUCID model or the USP28ΔUCID structure, the standard *Structure Preparation Module* of MOE was used. Afterward, E366/E373 was changed to alanine to obtain the mutated systems.

Protonation states and orientation of hydrogens were determined using MOEs *protonate3D* tool. H261/H268 results as neutral in this protocol, with the hydrogen at the ε-nitrogen (HIE).

These structures were initially minimized for 2000 steps to relax the system to the ff14SB force field using *pmemd* of Amber22 (Case DA et al, 2023) and an implicit solvent model (Onufriev et al, 2004). A box of TIP3P water molecules, measuring 10 Å from the solute to the box wall in each direction, was used to solvate all proteins. Given that USP25 and USP28 are highly negatively charged proteins, special attention was paid to the proper amount of ions in the simulation box to correctly mimic a salt concentration of 0.15 M. For this reason, the protocol suggested by Machado et al was followed (Machado and Pantano, 2020). For the 16-fold negatively charged USP25 system, 66 sodium cations and 50 chloride anions were added, while for the 18-fold negatively charged USP28 system, 62 sodium cations and 44 chloride anions were added (as the USP28 simulation box is slightly smaller than the USP25 box, fewer ions were needed). For the mutated versions, one anion less was employed. Freezing the protein structure, the solvent was then minimized for 2000 steps as well.

Starting from these systems, six independent equilibrations and MD simulations were performed for each. For every step of the MD protocol, *pmemd* of Amber22 was used with a time step of 2 fs, Langevin dynamics with a collision frequency of 3 ps$^{-1}$ as well as a cutoff of 10 Å for the particle-mesh Ewald summation. For the equilibration protocol, a stepwise procedure was chosen: After initially heating up the solvent from 100 to 300 K within 500 ps while simultaneously restraining the protein with a harmonic potential with a force constant of 100 kcal/mol*Å$^2$, the system was cooled down again to 100 K within 500 ps. In this step, the restraining force constant was lowered to 10 kcal/mol*Å$^2$, to slowly increase the mobility of the protein without simultaneously heating the solvent. In the final temperature step, the system was heated back to 300 K within 500 ps, using a force constant of 1 kcal/mol*Å$^2$ on the protein. These steps were performed in the NVT ensemble. Afterward, a 1 ns NPT simulation without any restraints was pursued, to adjust the density of the box. In total, 250 ns of production run were then performed for each replicate, saving a frame every 10 ps, resulting in a total sampling of 1.5 μs for each system.

For the analysis of the obtained trajectories, the AmberTool *cpptraj* was used. After aligning every frame onto the backbone of two adjacent β-sheets (L363/L370-F370/F377 and Y643/Y650-N649/N656), all frames were clustered using a hierarchical agglomerative clustering approach. The backbone RMSD of the N-terminal amino acids of helix α5 (S257/S264-W266/W273) was used as a clustering criterion. An additional principal component analysis (PCA) is described in Fig. EV5C.

## Funding

H2020 Societal Challenges EUbOPEN grant 875510 to VHO. Open Access funding enabled and organized by Projekt DEAL.

## Data availability

The models of the USP28 apo and -inhibitor-bound structures as well as crystallographic data have been deposited to the Protein Data Bank under the entry codes: 8P19 (USP28cat ΔUCID apo) (https://www.rcsb.org/structure/8P19), 8P14 (USP28cat ΔUCID VSM) (https://www.rcsb.org/structure/8P14), 8P1P (USP28cat Δtip P280H–AZ1) https://www.rcsb.org/structure/8P1P and 8P1Q (USP28cat Δtip FT206) (https://www.rcsb.org/structure/8P1Q). Further raw data are available in the Source Data.

The source data of this paper are collected in the following database record: biostudies:S-SCDT-10_1038-S44319-024-00167-w.

## Peer review information

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

## Acknowledgements

This work was supported by the Deutsche Forschungsgemeinschaft through FOR2134 (CK) and GRK2243 (CK and CS). We would like to thank David Andrews (AstraZeneca, Cambridge, UK) for providing AZ1. Diffraction data were collected at beamlines BL14.1 (BESSY II, HZB, Berlin, Germany), ID23-2 (ESRF, Grenoble, France), and P14 (PETRA III, EMBL/DESY, Hamburg, Germany). The authors would like to thank the beamline staff for assistance and support, Jochen Kuper (RVZ, Würzburg, Germany) for data collection on BL14.1 as well as Sebastian Bothe (RVZ, Würzburg, Germany) for support with in silico analysis of MD-data.

## Author contributions

**Jonathan Vincent Patzke**: Formal analysis; Validation; Investigation; Visualization; Writing—original draft; Writing—review and editing. **Florian Sauer**: Conceptualization; Formal analysis; Validation; Investigation; Visualization; Writing—original draft; Writing—review and editing. **Radhika Karal Nair**: Formal analysis; Investigation. **Erik Endres**: Software; Formal analysis; Investigation; Visualization; Writing—original draft. **Ewgenij Proschak**: Resources. **Victor Hernandez-Olmos**: Resources. **Christoph Sotriffer**: Formal analysis; Funding acquisition; Visualization; Writing—original draft; Writing—review and editing. **Caroline Kisker**: Conceptualization; Supervision; Funding acquisition; Writing—original draft; Writing—review and editing.

Source data underlying figure panels in this paper may have individual authorship assigned. Where available, figure panel/source data authorship is listed in the following database record: biostudies:S-SCDT-10_1038-S44319-024-00167-w.

## Disclosure and competing interests statement

The authors declare no competing interests.

# Expanded View Figures

**Figure EV1. USP28 and USP7-inhibitor complex structures (corresponds to main Fig. 1).**

(**A**) USP28-inhibitor-bound structures. Superposition of the complex structures of USP28 with AZ1 (blue), VSM (magenta) and FT206 (teal) and closeup view on the inhibitor-binding site. Helices α5 are highlighted in the same colors as the inhibitors. (**B**) USP7-inhibitor complex structures. Superposition of structures of USP7 apo (PDB 1NB8) (Hu et al, 2002; Data ref: Hu et al, 2003a), Ub-bound (PDB 1NBF) (Hu et al, 2002; Data ref: Hu et al, 2003b) and in complex with the thumb-palm cleft binding inhibitors Cpd2 (PDB 5WHC; bright green) (Di Lello et al, 2017; Data ref: Murray et al, 2017a), GNE6776 (Kategaya et al, 2017; Data ref: Murray et al, 2017b) and the catalytic channel binding inhibitors ALM2 (PDB 5N9R; dark green) (Gavory et al, 2018; Data ref: Harrison et al, 2017) and FT827 (PDB 5NGF; cyan) (Turnbull et al, 2017; Data ref: Krajewski et al, 2017). Helices corresponding to USP28 α4, α5 and α6 are colored in light red (apo), yellow (Ub-bound) or gray (inhibitor-bound), respectively. The thumb-palm cleft, the catalytic channel as well as the catalytic cysteine (C223) of USP7 are marked. The USP7 helix corresponding to the USP28 α5 moves upon Ub-binding towards the catalytic channel. This movement is blocked by the displayed inhibitors binding to the thumb-palm cleft or the catalytic channel. (**C**) Detailed views of the inhibitor-binding sites. AZ1 (left; blue), VSM (center; magenta) and FT206 (right; teal) are shown with the electron density maps of the inhibitors (at 1 σ). Residues of the thumb-palm cleft binding site are shown as sticks. Hydrogen bonds between the inhibitors and USP28 are displayed as dashed lines.

▶

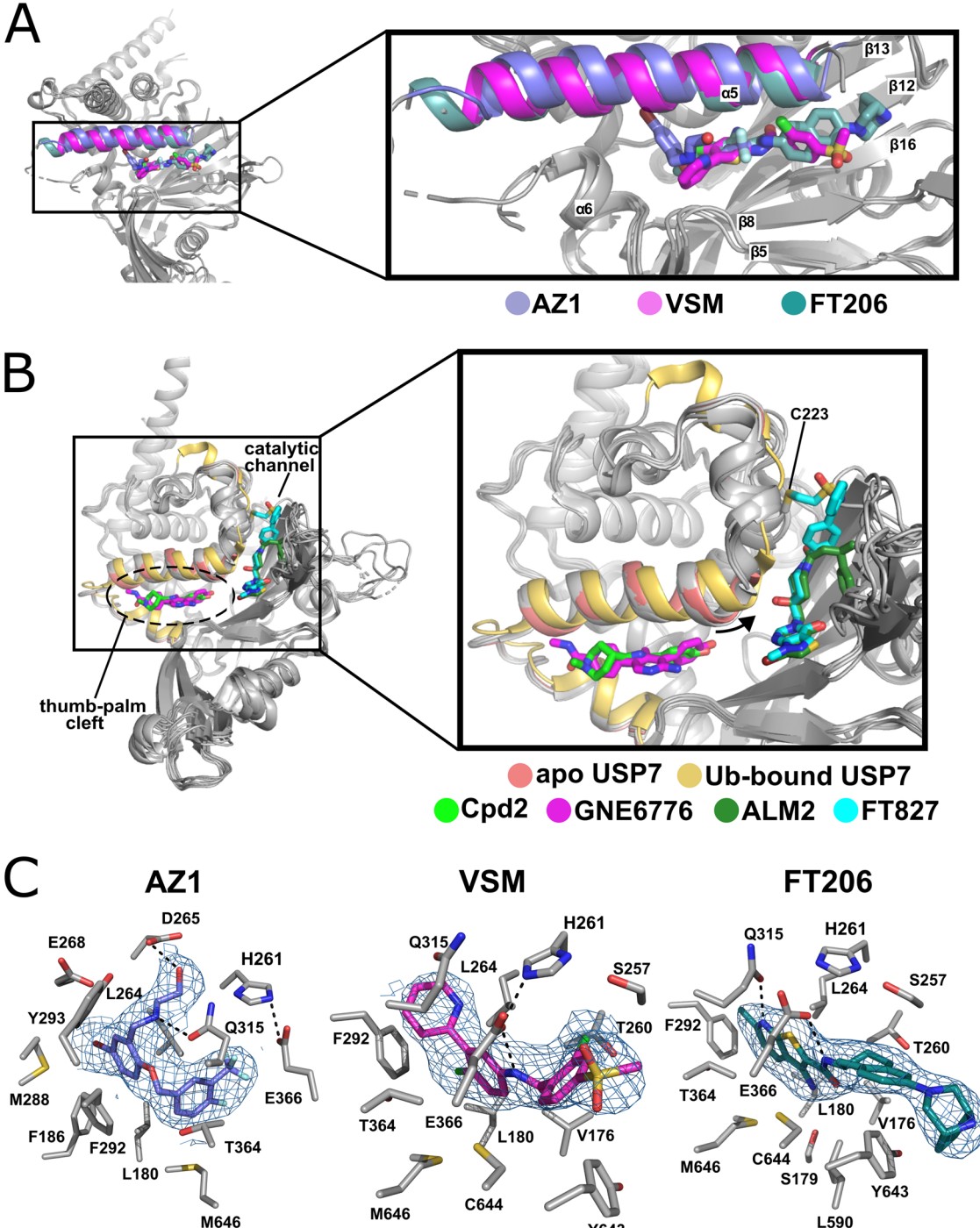

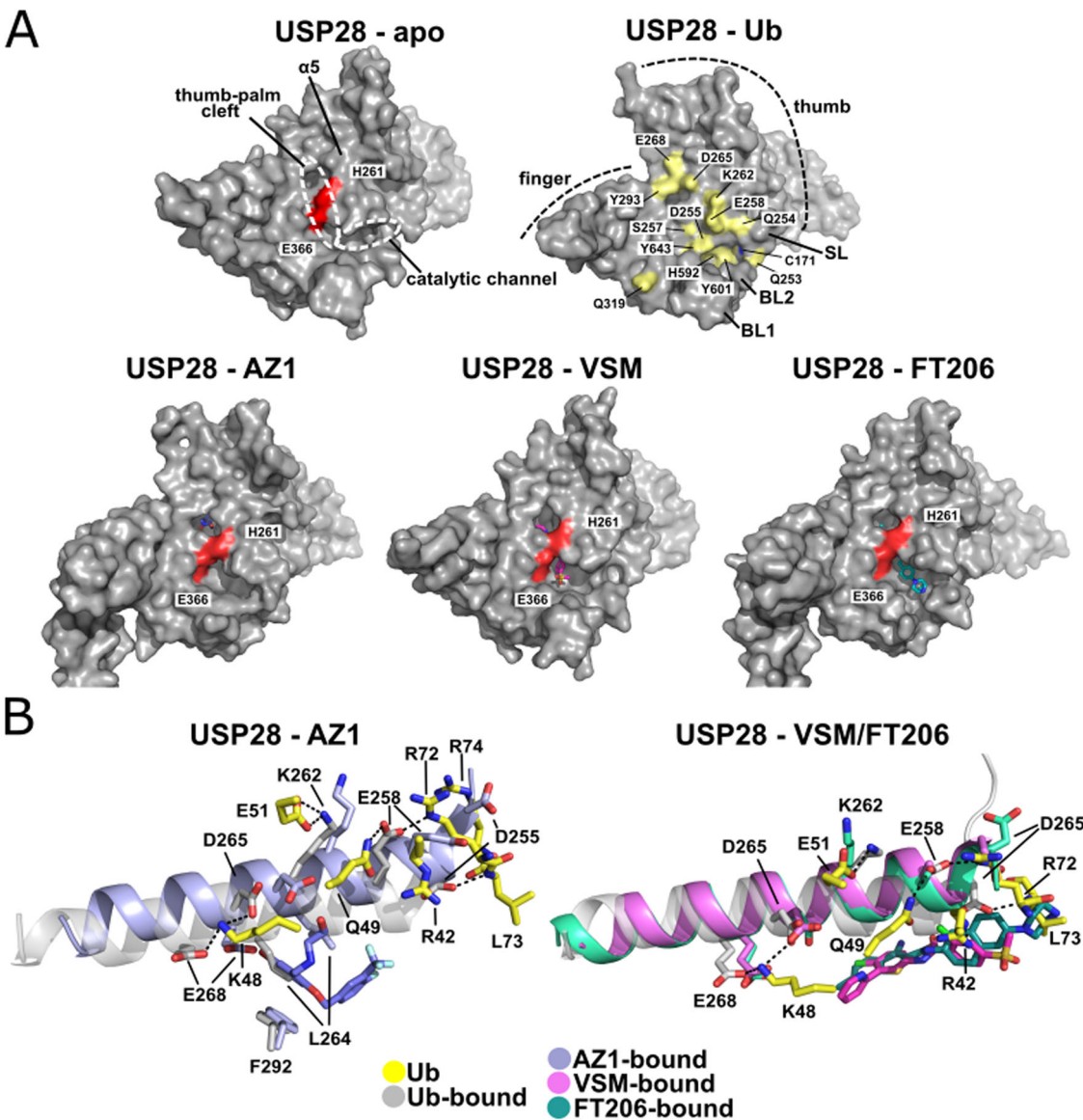

**Figure EV2. Inhibitory mechanism (corresponds to main Fig. 2).**

(A) Comparison of USP28 apo, Ub- and inhibitor-bound states. Top panel: Surface view of the USP28ΔUCID apo (this work) and Ub-bound (PDB 6HEI) (Gersch et al, 2019; Data ref: Gersch and Komander, 2019a) states. Transition from the apo- to the Ub-bound state is accompanied by the reshaping of the S1-site involving closure of the partially open thumb-palm cleft and a widening of the catalytic channel by movement of helix α5 (marked in the apo state). Most residues forming hydrogen bonds and salt bridges with Ub in the S1-site (marked yellow) cluster around the thumb-palm cleft and the catalytic channel. The catalytic cysteine (C171) is highlighted in blue. Bottom panel: Surface view on the inhibitor-bound states. Binding of AZ1, VSM and FT206 in the thumb-palm cleft locks the domain in an apo-like state, where the cleft cannot be closed. Side chains of H261 and E366 which lock the inhibitor-binding sites to the solvent, are highlighted in red. Note that in the apo- and inhibitor-bound states, some mobile elements of the catalytic channel (BL1, SL) are not completely modeled. It therefore appears to be more narrow in the Ub-bound state. (B) Closeup view on helix α5 of inhibitor-bound USP28. USP28Δtip P280H—AZ1 (blue, left panel), USP28ΔUCID wt—VSM (pink, right panel) and USP28Δtip wt— FT206 (teal, right panel) superimposed with Ub-bound USP28ΔUCID (PDB 6HEI) (Gersch et al, 2019; Data ref: Gersch and Komander, 2019a). Residues of helix α5 involved in hydrogen bonds and salt-bridge formation with Ub and corresponding positions in the inhibitor-bound state are shown as sticks and hydrogen bonds with Ub as dashed lines.

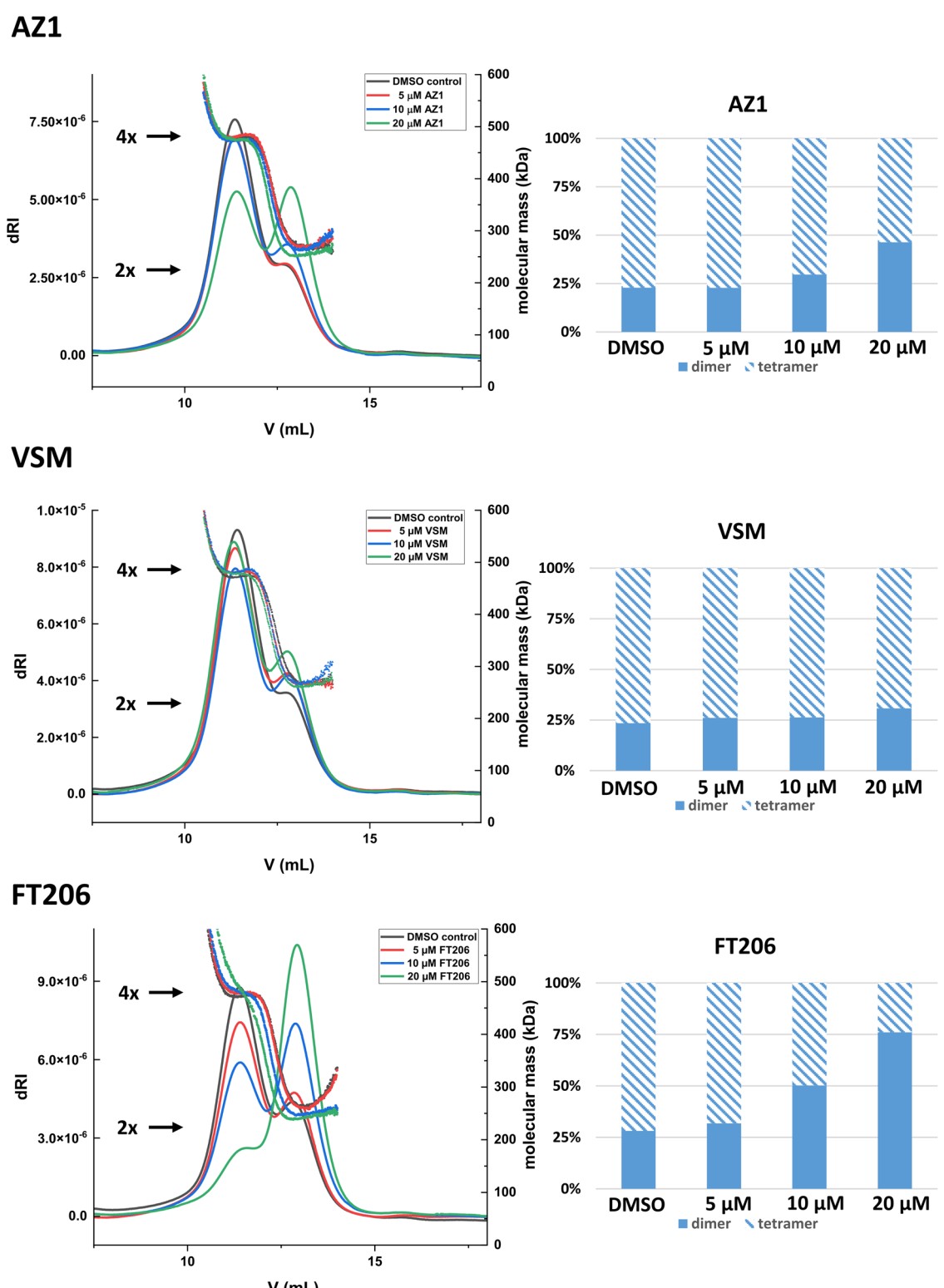

**Figure EV3. Inhibitor concentration-dependent dissociation of USP25 tetramers (corresponds to main Fig. 3).**

SEC-MALS analysis of 10 μM USP25fl after 45 min incubation with DMSO (black) or 5 μM (red), 10 μM (blue) and 20 μM (green) of AZ1, VSM or FT206 are displayed in the left panels from top to bottom, respectively. Continuous lines represent the protein concentration signal (refractive index, RI). Dots show the molecular mass calculated from RI and light scattering. Fractions of dimers and tetramers calculated from the average molecular mass of the entire peak fraction, each from a single experiment are represented in the bar diagram on the corresponding right panels. Source data are available online for this figure.

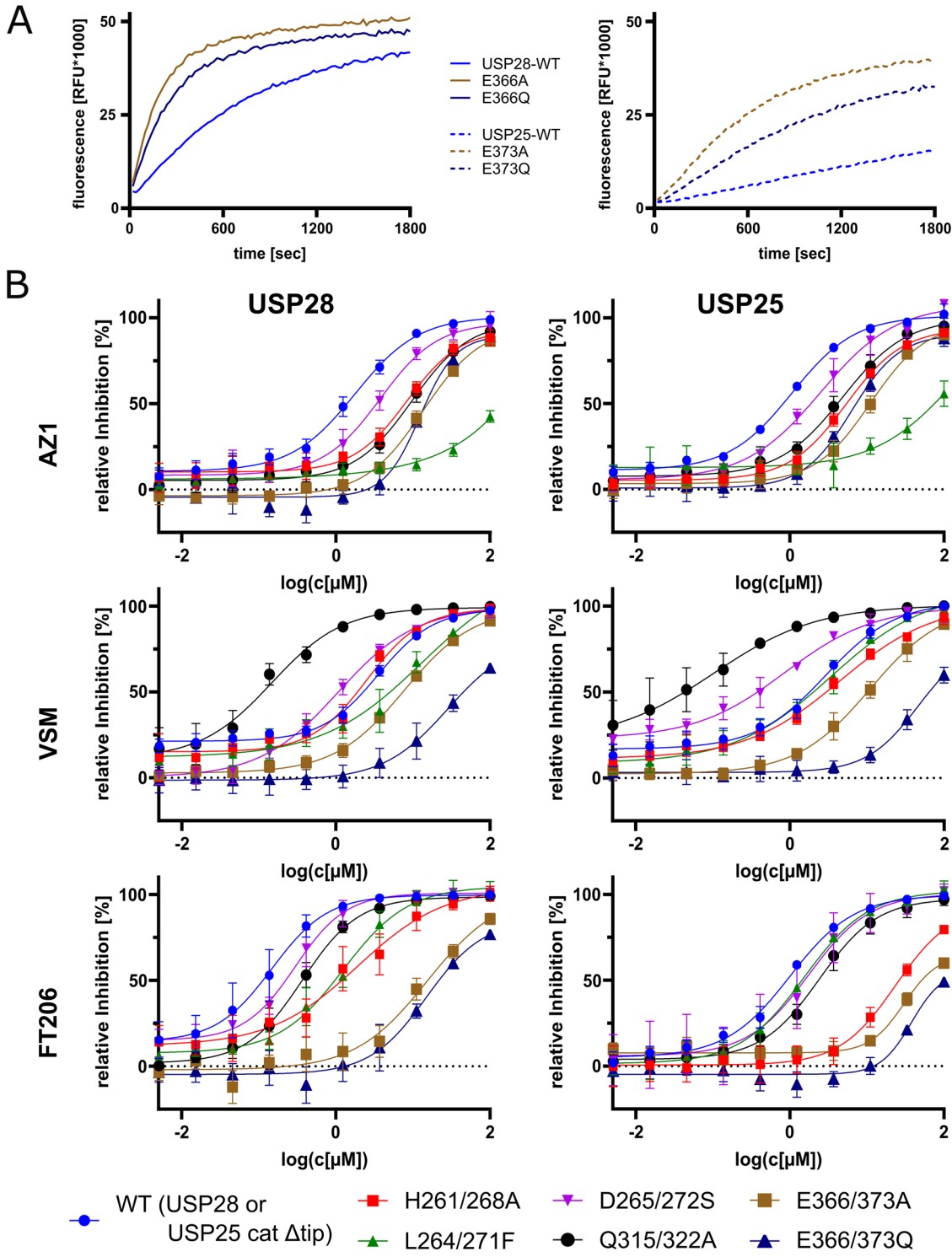

**Figure EV4.** (corresponds to main Fig. 5).

(A) Catalytic activity of wt and hyperactive USP28 and USP25 variants. Representative Ub-Rh110 cleavage assay of USP28Δtip (left panel) or USP25Δtip (right panel) variants (wt (blue), E366A/E373A (brown) and E366Q/E373Q (dark blue). The fluorescence signal (RFU*10³) is plotted against the time [s]. (B) Inhibitory potencies of AZ1, VSM and FT206. Non-linear regression of the dose–response assay for the different USP28 (left) and USP25 (right panels) Δtip variants with AZ1, VSM and FT206, from top to bottom, respectively. Dots represent the mean ± SD ($n \geq 5$, with two biological replicates) at corresponding inhibitor concentrations. The calculated IC50 values are depicted in Fig. 4C. Source data are available online for this figure.

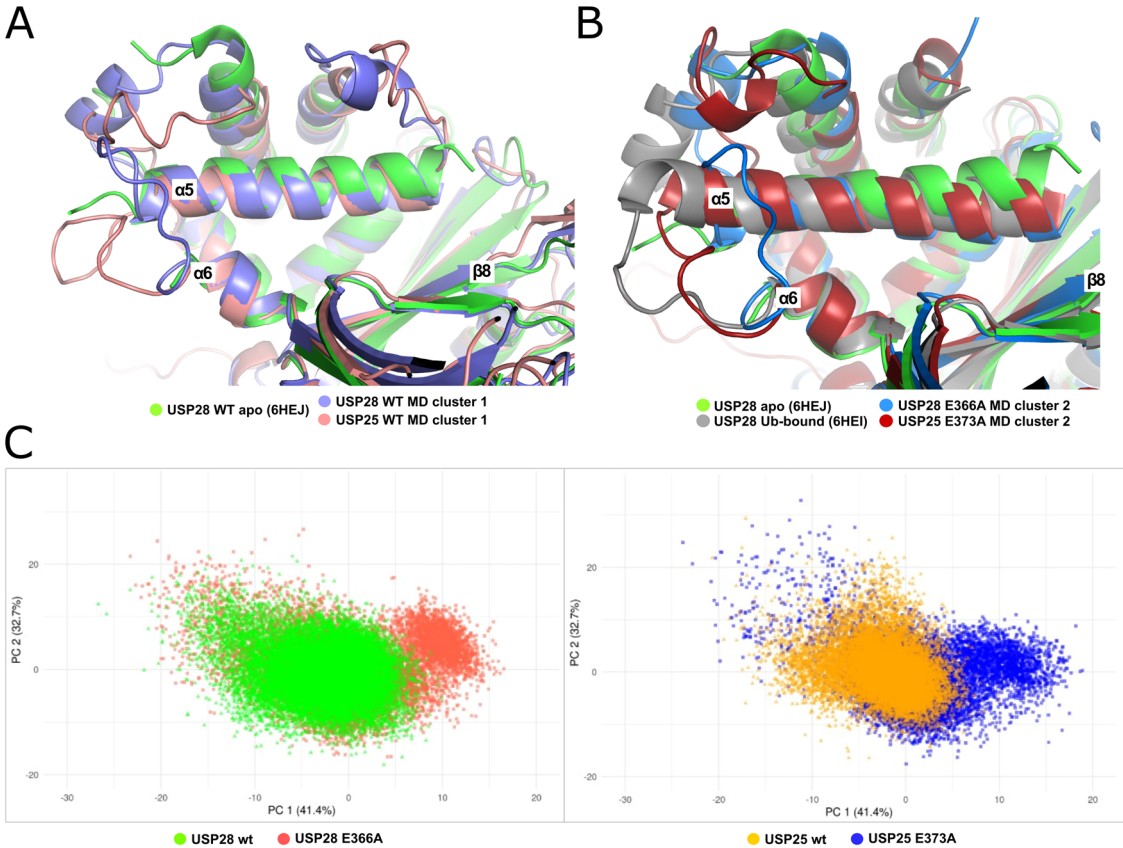

**Figure EV5.** (corresponds to main Fig. 6).

(A) Superposition of the representative snapshots of cluster 1 for USP25ΔUCID (pink) and USP28ΔUCID (blue) with the corresponding crystal structure of apo USP28ΔUCID (PDB 6HEJ; green) (Gersch et al, 2019; Data ref: Gersch and Komander, 2019b). Secondary structure elements containing residues mediating bonds with inhibitor (α5, α6 and β8) are marked. (B) Superposition of the crystal structure of apo USP28ΔUCID (PDB 6HEJ; green) (Gersch et al, 2019; Data ref: Gersch and Komander, 2019b) and Ub-bound (PDB 6HEI; gray) (Gersch et al, 2019; Data ref: Gersch and Komander, 2019a) with representative snapshots of cluster 2 for USP28ΔUCID E366A (blue) and USP25ΔUCID E373A (red). Secondary structure elements containing residues mediating bonds with inhibitor (α5, α6, β5 and β8) are marked. (C) Two-dimensional principal subspace displaying the differences between wt and E366A/E373A of USP28 (left) and USP25 ΔUCID variants (right panel). For the initial PCA one frame every 100 ps was utilized for the MD trajectories of USP28ΔUCID E366A and the eigenvectors were constructed based on the ten N-terminal backbone atoms of helix α5 and an alignment onto two adjacent β-strands (L363-F370 and Y643-N649). The first two principal components PC1 and PC2 together capture about 74% of the variance within these coordinates (red). All further PCA calculations, both for USP28 wt (green), USP25 wt (yellow) and E373A (blue) were performed utilizing the same alignment and atom selection as described above. The coordinates were expressed in terms of PC1 and PC2 of USP28 E366A to ensure direct comparability between both wt and E366A/E373A as well as USP25 and USP28. Source data are available online for this figure.

