## [Peer Review File · EMBO Reports]

Structural basis for the bi-specificity of USP25 and USP28 inhibitors

Jonathan Vincent Patzke, Florian Sauer, Radhika Karal Nair, Erik Endres, Ewgenij Proschak, Victor Hernandez-Olmos, Christoph Sotriffer and Caroline Kisker

Corresponding author(s): Caroline Kisker (caroline.kisker@virchow.uni-wuerzburg.de)

Review Timeline:

Submission Date:	13th Nov 23
Editorial Decision:	12th Jan 24
Revision Received:	5th Apr 24
Editorial Decision:	6th May 24
Revision Received:	10th May 24
Accepted:	13th May 24

Transaction Report:

Dear Prof. Kisker

Thank you for the submission of your research manuscript to our journal. I apologize for the delay in handling your manuscript, but we have only now received the full set of referee reports that is copied below.

As you will see, the referees acknowledge that the findings are potentially interesting, but they also raise a number of concerns that will need to be addressed in a revised version.

Given the constructive comments and support by the referees, we would like to invite you to revise your manuscript with the understanding that the referee concerns (as detailed above and in their reports) must be fully addressed and their suggestions taken on board. Please address all referee concerns in a complete point-by-point response. Acceptance of the manuscript will depend on a positive outcome of a second round of review. It is EMBO Reports policy to allow a single round of revision only and acceptance or rejection of the manuscript will therefore depend on the completeness of your responses included in the next, final version of the manuscript.

We realize that it is difficult to revise to a specific deadline. In the interest of protecting the conceptual advance provided by the work, we recommend a revision within 3 months (April 12th). Please discuss the revision progress ahead of this time with the editor if you require more time to complete the revisions.

I am also happy to discuss the revision further via e-mail or a video call, if you wish.

*****IMPORTANT GENERAL NOTE:

We perform an initial quality control of all revised manuscripts before re-review. Your manuscript will FAIL this control and the handling will be delayed IN CASE the following APPLIES:

- 1) If a data availability section providing access to data deposited in public databases is missing. If you have not deposited any data, please add a sentence to the data availability section that explains that.
- 2) If your manuscript contains statistics and error bars based on $n=2$. Please use scatter blots in these cases. No statistics should be calculated if $n=2$.

When submitting your revised manuscript, please carefully review the instructions that follow below. Failure to include requested items will delay the evaluation of your revision.*****

- 1) a .docx formatted version of the manuscript text (including legends for main figures, EV figures and tables). Please make sure that the changes are highlighted to be clearly visible.
- 2) individual production quality figure files as .eps, .tif, .jpg (one file per figure). Please download our Figure Preparation Guidelines (figure preparation pdf) from our Author Guidelines pages <https://www.embopress.org/page/journal/14693178/authorguide> for more info on how to prepare your figures.
- 3) a .docx formatted letter INCLUDING the reviewers' reports and your detailed point-by-point responses to their comments. As part of the EMBO Press transparent editorial process, the point-by-point response is part of the Review Process File (RPF), which will be published alongside your paper.
- 4) a complete author checklist, which you can download from our author guidelines (<<https://www.embopress.org/page/journal/14693178/authorguide>>). Please insert information in the checklist that is also reflected in the manuscript. The completed author checklist will also be part of the RPF.
- 5) Please note that all corresponding authors are required to supply an ORCID ID for their name upon submission of a revised manuscript (<<https://orcid.org/>>). Please find instructions on how to link your ORCID ID to your account in our manuscript tracking system in our Author guidelines (<<https://www.embopress.org/page/journal/14693178/authorguide#authorshipguidelines>>)
- 6) We replaced Supplementary Information with Expanded View (EV) Figures and Tables that are collapsible/expandable online. A maximum of 5 EV Figures can be typeset. EV Figures should be cited as "Figure EV1, Figure EV2" etc... in the text and their

respective legends should be included in the main text after the legends of regular figures.

7) Before submitting your revision, primary datasets (and computer code, where appropriate) produced in this study need to be deposited in an appropriate public database (see <<https://www.embopress.org/page/journal/14693178/authorguide#dataavailability>>). Specifically, we would kindly ask you to provide public access to the structural data.

Please remember to provide a reviewer password if the datasets are not yet public and if the database provides this option.

The accession numbers and database should be listed in a formal "Data Availability " section (placed after Materials & Method) that follows the model below (see also <<https://www.embopress.org/page/journal/14693178/authorguide#dataavailability>>). Please note that the Data Availability Section is restricted to new primary data that are part of this study.

Data availability

Additional information on source data and instruction on how to label the files are available <<https://www.embopress.org/page/journal/14693178/authorguide#sourcedata>>.

10) Figure legends and data quantification:

- the name of the statistical test used to generate error bars and P values,
- the number (n) of independent experiments (please specify technical or biological replicates) underlying each data point,
- the nature of the bars and error bars (s.d., s.e.m.)
- If the data are obtained from n {less than or equal to} 5, show the individual data points in addition to the SD or SEM.
- If the data are obtained from n {less than or equal to} 2, use scatter blots showing the individual data points.

11) Our journal encourages inclusion of *data citations in the reference list* to directly cite datasets that were re-used and obtained from public databases. Data citations in the article text are distinct from normal bibliographical citations and should directly link to the database records from which the data can be accessed. In the main text, data citations are formatted as follows: "Data ref: Smith et al, 2001" or "Data ref: NCBI Sequence Read Archive PRJNA342805, 2017". In the Reference list,

data citations must be labeled with "[DATASET]". A data reference must provide the database name, accession number/identifiers and a resolvable link to the landing page from which the data can be accessed at the end of the reference. Further instructions are available at <<https://www.embopress.org/page/journal/14693178/authorguide#referencesformat>>.

12) All Materials and Methods need to be described in the main text. We would encourage you to use 'Structured Methods', our new Materials and Methods format. According to this format, the Materials and Methods section should include a Reagents and Tools Table (listing key reagents, experimental models, software and relevant equipment and including their sources and relevant identifiers) followed by a Methods and Protocols section in which we encourage the authors to describe their methods using a step-by-step protocol format with bullet points, to facilitate the adoption of the methodologies across labs. More information on how to adhere to this format as well as downloadable templates (.doc or .xls) for the Reagents and Tools Table can be found in our author guidelines: <<https://www.embopress.org/page/journal/14693178/authorguide#manuscriptpreparation>>. An example of a Method paper with Structured Methods can be found here: <<https://www.embopress.org/doi/10.15252/msb.20178071>>.

13) As part of the EMBO publication's Transparent Editorial Process, EMBO Reports publishes online a Review Process File to accompany accepted manuscripts. This File will be published in conjunction with your paper and will include the referee reports, your point-by-point response and all pertinent correspondence relating to the manuscript.

Yours sincerely,

Referee #1:

USP25 and USP28 are two closely related enzymes with virtually identical active sites. Hence identifying specific inhibitors is hard and this is exemplified by recent HTS campaigns identifying several USP28 inhibitors that cross-react and inhibit USP28 with similar IC50s.

Patzke, Saur et al., have set out to identify the structural and mechanistic rules for inhibitor cross-reactivity and put forward ideas for how to identify more specific inhibitors in the future.

The authors have used two different conditions to obtain crystals that are amenable for soaking. They managed to solve USP28 co-structures with AZ1, VSM and FT206.

All three inhibitor-bound structures revealed that small molecules inhibit USP28 by a common allosteric mechanism consistent with previously-published data. It is nice to see this confirmed with co-structures of USP28-bound to inhibitors thus showing the precise allosteric mechanisms.

Also, inhibitors bind similarly to a hydrophobic cleft (same cleft as USP25 auto-inhibitory UCID tip) and affect dimer - tetramer states. This is a very interesting insight.

Structures and inhibitor binding sites are validated by mutagenesis and sequence similarity analysis with other USPs. This has also provided some interesting and valuable information for inhibitor cross-reactivity between USP25 and USP28, and inhibitor specificity over other DUBs.

Overall, the study is rigorous and has very important implications for the discovery of selective USP25/USP28 inhibitors as well as DUB inhibitors in general. As such, this will be of interest to a wide range of academic and industry-facing audiences.

If the authors are invited to submit a revised version, I have the following minor comments that need to be addressed before publication is warranted.

Fig 1B. Label panels to identify inhibitors. Why order for VSM and FT206 panels are not consistent with panels in A?

Fig. 2B. Would be good to show how VSM / FT206 relate to USP25-Ub structure.

Electron density: unbiased fo-fc maps at $\sim 2.5\sigma$ (i.e. maps before the compound was built and refined with the protein model). If these are not available the authors should show omit maps after simulated annealing.

The authors use USP28 L264F and USP25 L271F variants to analyze the impact of the stacking interaction between AZ1 and F292/F299, since this locks the AZ1 bromophenyl binding pocket through Phe-Phe stacking. It is hard to appreciate this point without referring to a figure.

A cartoon model is required to aid discussion and explain the mechanism of inhibition. It is also important to highlight the allosteric mechanism of inhibition and its impact on dimer - tetramer states in the model.

Referee #2:

Patzke et al. present crystal structures of known small molecule inhibitors (AZ1, VSM and FT206) in complex with the USP28 catalytic domain that have been shown to exhibit similar potency for USP28 and its paralogue USP25. Despite being structurally distinct, the authors show that the three inhibitors bind to overlapping positions in a USP28 hydrophobic cleft of the S1 distal ubiquitin binding site between the thumb and palm of the USP fold. Interestingly, the region overlaps with the autoinhibitory loop binding site in paralogue USP25. The importance of selected individual residues to the inhibition was then probed for both paralogues by using binding site mutants and the measurement of dose-response curves. This revealed that inhibitors likely adopt the same binding modes in USP28 and USP25 and provides insight into their contribution to the inhibition. Sequence conservation of contributing residues explains the lack of selectivity between the paralogues and provides a rationale for why other USPs are not targeted by these inhibitors. They then focus on USP28 E366 (USP25 E373), which they identify as a main common facilitator residue for inhibition of both USPs by the selected inhibitors and further evaluate this residue's role in inhibition and its increased catalytic activity when mutated to an alanine compared to WT in MD simulations. The original data provide novel insights into why the inhibitors do not discriminate between the two paralogues and do not act on other members of the USP family, which will be of interest to the field. Structures of USPs in complex with small molecule inhibitors are still relatively rare, many USPs key therapeutic targets, and the data presented clearly adds to the knowledge base. However, there are some issues that need resolving.

Key points

- The manuscript is generally well-written but wordy with some key points getting lost in detail, which could be better and more concisely explained if more and better labelled figures were included (see below). This should be addressed in a revised version. Also, there is a lot of emphasis in the introduction/discussion on the role of USP28 and USP25 in cancer, yet this isn't the main focus of the manuscript, so this should be shortened. Instead, more room should be given to modes of inhibition of other USPs by small molecule inhibitors, where structural information is available and how the findings compare.
- Abstract: should provide more detail: Name all three inhibitors investigated. "Importantly, we identified a key residue in the

inhibitor binding site which facilitates the inhibition by all residues." Should this read by all three inhibitors tested, and is this strictly true? Please clarify statement. It also doesn't seem to be fully backed up by the reported IC50 values in the table. The key residue identified should be named in the abstract. This is also repeated in the results section, is this justified? The effects of the E366A/E373A mutants are not equal for the different inhibitors; the mutant only had a weak effect on the inhibition by VSM, moderate for AZ1 and only strongly for FT206, which is not in line with the statement that E366/E373 had a significant impact on inhibition by all inhibitors, please provide a more nuanced statement.

- Discussion: elaborate on how new insights complement previous experiments by Wang et al. to characterise VSM binding.
- Table 1: average B factors for inhibitors in the structures need to be given in separation (not combined with other ligand/ions) so that occupancy/fit of the inhibitors can be gaged.
- The clarity of figures needs improvement.
 - A figure for the chemical structures of the three inhibitors investigated should be included.
 - To demonstrate that the three inhibitors share the same binding site a superposition of the three structures should be shown, so that the extent of overlap of the binding modes is apparent.
 - Equally, Ligplot figures or equivalent should be included to show all interactions (hydrophobic as well as H-bonding) with the inhibitors. This will improve the clarity and provide a more detailed picture of all interactions (and the rationale for selecting mutations).
 - Labelling needs improvement in most figures:
 - Fig. 1 A: please label sec. structure elements in all structures and highlight catalytic triad residues for orientation; change from yellow to a colour that is better visible; make sure label of inhibitory tip points directly to the loop in green rather than another loop in the structure.
 - Fig. 1B: label sec. structure elements the residues belong to and include leucine 264, which was chosen for mutagenesis.
 - Fig 4B: don't cut off structure figure and clearly label residues highlighted in yellow.
 - Fig. 5A: individual data points should be shown in the bar charts and is the error displayed SEM or SD?
 - Fig. 5C: Leu264 and Asp265 should be included to illustrate why these have been chosen for mutagenesis.
 - Fig. 6A: label key structural elements in the figure.
 - A comparison with other USP-inhibitor structures, where the inhibitors also bind to the palm thumb cleft should be provided as a figure (superposition or side by side).
 - Fig EV2A: label residues highlighted in yellow; clearly label position of E366 and H261 in figures.
 - Fig. EV5: label key sec. structure elements

Referee #3:

This is a nice study describing X-ray co-structures of three chemically distinct dual USP25/28 inhibitors to USP28 accompanied by biophysical and biochemical studies using wild-type and mutant USP25 and USP28 enzymes to characterize key determinants of binding and inhibition as well as the effect of compound binding on enzyme dimers and tetramers. The data combine for an excellent story providing novel insight into USP25/28 inhibition as well as DUBs more generally. Studies are logical, rigorous and well described. The manuscript is recommended for publication following some minor edits/clarifications.

- Need to clearly define S1. In DUB space, this refers to distal ubiquitin binding site where as in general protease literature it refers to the binding cleft for the P1 amino acid of the substrate.
- The current text in both the results and discussion sections suggests that the USP1 inhibitor ML323 binds in the same cleft as the USP28 inhibitors and USP7 inhibitors. The USP1 inhibitor ML323 binds in a cryptic pocket on the other side of the DUB domain. This needs to be clear.
- Since USP25 exists in an autoinhibited state, it has been shown to be less active in biochemical assays than USP28. A comment on how this impacts comparing the IC50s for the two enzymes to each other will be useful.

Response to Reviewers' Comments

We thank the reviewers for their constructive comments and appreciate their significant and timely efforts providing helpful feedback to improve our manuscript. The revised manuscript includes a modifications in the text as well as modified Figures and new subfigures for better illustration.

Referee #1:

USP25 and USP28 are two closely related enzymes with virtually identical active sites. Hence identifying specific inhibitors is hard and this is exemplified by recent HTS campaigns identifying several USP28 inhibitors that cross-react and inhibit USP28 with similar IC50s.

Patzke, Saur et al., have set out to identify the structural and mechanistic rules for inhibitor cross-reactivity and put forward ideas for how to identify more specific inhibitors in the future.

The authors have used two different conditions to obtain crystals that are amenable for soaking. They managed to solve USP28 co-structures with AZ1, VSM and FT206.

All three inhibitor-bound structures revealed that small molecules inhibit USP28 by a common allosteric mechanism consistent with previously-published data. It is nice to see this confirmed with co-structures of USP28-bound to inhibitors thus showing the precise allosteric mechanisms.

Also, inhibitors bind similarly to a hydrophobic cleft (same cleft as USP25 auto-inhibitory UCID tip) and affect dimer - tetramer states. This is a very interesting insight.

Structures and inhibitor binding sites are validated by mutagenesis and sequence similarity analysis with other USPs. This has also provided some interesting and valuable information for inhibitor cross-reactivity between USP25 and USP28, and inhibitor specificity over other DUBs.

Overall, the study is rigorous and has very important implications for the discovery of selective USP25/USP28 inhibitors as well as DUB inhibitors in general. As such, this will be of interest to a wide range of academic and industry-facing audiences.

If the authors are invited to submit a revised version, I have the following minor comments that need to be addressed before publication is warranted.

Fig 1B. Label panels to identify inhibitors. Why order for VSM and FT206 panels are not consistent with panels in A?

The reviewer is correct and this was an oversight. The order of AZ1, VSM and FT206 has been adjusted, please see Fig. 1B.

Fig. 2B. Would be good to show how VSM / FT206 relate to USP25-Ub structure.

So far no structure of an apo or Ub-bound form of the USP25 catalytic domain has been reported. The currently only available structure of USP25 is of its autoinhibited homo-tetramer (PDB 5O71/6H4J/6HEL). Due to their high structural similarity and similar, while not identical, behavior in biochemical assays (Sauer *et al*, 2019; Gersch *et al*, 2019), our currently best model for how the inhibitors bind to USP25 is USP28. In Fig. 1A we compare the positions of the three inhibitors' binding sites with the position of the distal Ub bound to the USP28 S1 site and of the thumb-palm cleft bound portion of the USP25-tip loop. In Fig. 2B/EV2B we compare the Ub- and inhibitor bound states of USP28 to highlight the movements of the α 5-helix and residues directly involved in Ub binding.

Electron density: unbiased fo-fc maps at $\sim 2.5\sigma$ (i.e. maps before the compound was built and refined with the protein model). If these are not available the authors should show omit maps after simulated annealing.

Unbiased maps from before inhibitor modeling were not available as the remodeling of the protein due to structural alterations and modeling of the inhibitors was done simultaneously. We have therefore generated a set of simulated annealing composite omit maps using the pure protein models without inhibitors. These maps are shown in Appendix Fig. 1A.

The authors use USP28 L264F and USP25 L271F variants to analyze the impact of the stacking interaction between AZ1 and F292/F299, since this locks the AZ1 bromophenyl binding pocket through Phe-Phe stacking. It is hard to appreciate this point without referring to a figure.

We have improved Fig. 1C, now comprising the closeup view of the inhibitor binding sites, with the sidechain of L264, pointing into the thumb-palm-cleft to clarify the performed L264F/L271F mutagenesis.

A cartoon model is required to aid discussion and explain the mechanism of inhibition. It is also important to highlight the allosteric mechanism of inhibition and its impact on dimer - tetramer states in the model.

The cartoon model is certainly a very good idea. We addressed this in the revised version of the manuscript, and included this in the graphical abstract which schematically describes our key findings. The movement of the thumb especially $\alpha 5$ responsible for the open and closed conformation of the cleft, which allows either binding of the inhibitors to the thumb-palm cleft or ubiquitin to the S1-site.

Referee #2:

Patzke et al. present crystal structures of known small molecule inhibitors (AZ1, VSM and FT206) in complex with the USP28 catalytic domain that have been shown to exhibit similar potency for USP28 and its paralogue USP25. Despite being structurally distinct, the authors show that the three inhibitors bind to overlapping positions in a USP28 hydrophobic cleft of the S1 distal ubiquitin binding site between the thumb and palm of the USP fold. Interestingly, the region overlaps with the autoinhibitory loop binding site in paralogue USP25. The importance of selected individual residues to the inhibition was then probed for both paralogues by using binding site mutants and the measurement of dose-response curves. This revealed that inhibitors likely adopt the same binding modes in USP28 and USP25 and provides insight into their contribution to the inhibition. Sequence conservation of contributing residues explains the lack of selectivity between the paralogues and provides a rationale for why other USPs are not targeted by these inhibitors. They then focus on USP28 E366 (USP25 E373), which they identify as a main common facilitator residue for inhibition of both USPs by the selected inhibitors and further evaluate this residue's role in inhibition and its increased catalytic activity when mutated to an alanine compared to WT in MD simulations.

The original data provide novel insights into why the inhibitors do not discriminate between the two paralogues and do not act on other members of the USP family, which will be of interest to the field. Structures of USPs in complex with small molecule inhibitors are still relatively rare, many USPs key therapeutic targets, and the data presented clearly adds to the knowledge base. However, there are some issues that need resolving.

Key points

- The manuscript is generally well-written but wordy with some key points getting lost in detail, which could be better and more concisely explained if more and better labelled figures were included (see below). This should be addressed in a revised version. Also, there is a lot of emphasis in the introduction/discussion on the role of USP28 and USP25 in cancer, yet this isn't the main focus of the manuscript, so this should be shortened. Instead, more room should be given to modes of inhibition of other USPs by small molecule inhibitors, where structural information is available and how the findings compare.

We thank the reviewer for his comments. We carefully revised the manuscript and addressed the issues raised. In the introduction we had to keep some of the cancer related statements to explain why there is such a strong focus on the development of USP28 and USP25 inhibitors. In addition, we had to explain why bispecificity of inhibitors may be a serious obstacle for clinical use. The discussion has been rewritten to set the focus on how the here described inhibitors compare to other structurally characterized USP-inhibitors particularly those targeting USP7 where compounds binding into the thumb-palm cleft are known and structural rearrangements similar in extend to these seen in USP28 have been described. We also included an additional Subfigure for comparison (Fig. EV1B).

- Abstract: should provide more detail: Name all three inhibitors investigated.

The abstract has been changed accordingly and now includes the three compounds.

“Importantly, we identified a key residue in the inhibitor binding site which facilitates the inhibition by all residues.”

Should this read by all three inhibitors tested, and is this strictly true? Please clarify statement. It also doesn't seem to be fully backed up by the reported IC50 values in the table. The key residue identified should be named in the abstract. This is also repeated in the results section, is this justified? The effects of the E366A/E373A mutants are not equal for the different inhibitors; the mutant only had a weak effect on the inhibition by VSM, moderate for AZ1 and only strongly for FT206, which is not in line with the statement that E366/E373 had a significant impact on inhibition by all inhibitors, please provide a more nuanced statement.

We thank the reviewer for pointing out that we did not clearly explain our observations on E366A/E373A. and we agree with the reviewer that the statement doesn't seem to be fully backed by all E366/E373 variants and for all inhibitors, particularly for the alanine variants tested against VSM. In the revised version of the manuscript we describe this more thoroughly and with respect to the different inhibitors.

The discrepancy between weak effects of the E366/373 to alanine variants observed for VSM but moderate to strong effects observed for AZ1 and FT206 is now explained in different sections of the manuscript. In short they can be explained by the negative effect E366/E373 has on the binding of VSM through its interaction with Q315/Q322.

This is described in the section **“Molecular basis for bi-specificity and selectivity”**:

“In line with previous biochemical data (Wang et al, 2021a), mutation of Q315 in USP28 or Q322 in USP25 to alanine strongly increased the potency of VSM. Based on docking studies, this effect had been attributed to an outward flipping of the Q315 side chain that is required to accommodate VSM in the cleft. Our structures show that, in comparison to apo USP28, indeed such a flip takes place to provide sufficient space for the central chlorobenzene and pyridine rings of VSM (Fig. 5C). Similar movements of the glutamine side chain are also required to accommodate the FT206 thienopyridine and the AZ1 fluorine-substituted ring. However, for both compounds, the Q to A mutation did not

result in a strong increase but a moderate decrease in potency in either protein. A possible explanation for this can be found in the intramolecular hydrogen bond formed between the Q315 and E366 side chains in the USP28 apo state. This hydrogen bond needs to be broken to permit the Q315 flip. For FT206 and AZ1, the energetic costs for this are at least partially compensated by subsequent formation of a hydrogen bond with the compounds' thienopyridine respectively ethanamine nitrogen atoms, while this is not the case for VSM, where the glutamine remains unbound (Fig. 5C)."

Stronger, albeit not equally strong effects were observed for the corresponding E366/E373 to glutamine variants. This has been described in the same section:

"In contrast, release of the steric block between E366/E373 and Q315/Q322 may compensate for the loss of the hydrogen bond between E366/E373 and VSM. To further decipher the role of E366/E373, we generated the additional variants E366Q/E373Q which lack the negative charge of the glutamate side chain, while retaining the ability to bind Q315/Q322 through H-bonds. These variants displayed strongly increased catalytic activity and impairment of AZ1 and FT206 inhibition comparable to their alanine counterparts. However, both glutamine variants also suppressed inhibition by VSM to a similar extent as seen for the other inhibitors, thus emphasizing the importance of the negative charge provided by the E366/E373 carboxyl group in this position for inhibition."

Furthermore, as evidenced by the strong increase in catalytic activity observed for all E366/E373 variants, we identified a structural function of these glutamates as supported by the MD-simulations described in the section "**Role of glutamate 366/373**".

To explain this we included in the **discussion** the sentence: 'Importantly, we identified a key glutamate residue at positions 366/373 in USP28/USP25 in the binding site which plays a central structural role for pocket stability and thereby for inhibition and activity.'

Additionally, we discussed the individual roles of E366/E373 in a more detailed manner.

Taken together, we hypothesize in the **discussion** that individual differences such as number of formed bonds and conformational flexibility are the reasons why different compounds are affected differently. We discussed this in the revised manuscript: "The glutamate's effect in promoting a 'cleft open' state may further explain the observation of a stronger effect on FT206 binding, where a larger molecule forming only two H-bonds with the DUB has to be accommodated in the cleft, while a weaker effect is seen for the smaller AZ1 which may be more easily accommodated inside the cleft and forms more bonds with the DUB."

- Discussion: elaborate on how new insights complement previous experiments by Wang et al. to characterise VSM binding.

We now included a short part on binding site mapping by HDX-MS as well as mutational analysis by Wang et al. and docking attempts by the same group and Liu et al. (2020) in the discussion.

- Table 1: average B factors for inhibitors in the structures need to be given in separation (not combined with other ligand/ions) so that occupancy/fit of the inhibitors can be gaged.

We changed this accordingly, please see Table 1

- The clarity of figures needs improvement.

- A figure for the chemical structures of the three inhibitors investigated should be included.

We included a new top panel in Figure 1A, comprising the chemical structures of the three compounds AZ1, VSM and FT206.

- To demonstrate that the three inhibitors share the same binding site a superposition of the three structures should be shown, so that the extent of overlap of the binding modes is apparent.

The new subfigure EV1A (corresponds to main Fig. 1) contains a superposition of all three USP28-inhibitor complexes presented in this work together with a closeup view on the binding sites and the correspondingly colored $\alpha 5$ helices are shown.

- Equally, Ligplot figures or equivalent should be included to show all interactions (hydrophobic as well as H-bonding) with the inhibitors. This will improve the clarity and provide a more detailed picture of all interactions (and the rationale for selecting mutations).

Ligplot plots are now shown in the new Fig. S1C included in the Appendix. In addition, Fig. EV1C contains a detailed view on the binding site for each inhibitor.

- Labelling needs improvement in most figures:

- Fig.1 A: please label sec. structure elements in all structures and highlight catalytic triad residues for orientation; change from yellow to a colour that is better visible; make sure label of inhibitory tip points directly to the loop in green rather than another loop in the structure.

We have now included labels for $\alpha 1$, $\alpha 2$, $\alpha 5$, $\alpha 6$ as well as the catalytic triad in all overview subfigures for better clarity. Additionally, we have labelled $\beta 8$, $\beta 12$, $\beta 13$, $\beta 16$ in the inhibitor bound structures. The positioning of the inhibitory tip label and the color of ubiquitin have been adjusted.

- Fig. 1B: label sec. structure elements the residues belong to and include leucine 264, which was chosen for mutagenesis.

Secondary structure elements that belong to H261, L264 and D265 ($\alpha 5$), F292 ($\alpha 6$) and E366 ($\beta 8$) have been added. As Q315 locates on a linker between $\beta 4$ and $\beta 5$, we have labelled $\beta 4$ for orientation.

- Fig 4B: don't cut off structure figure and clearly label residues highlighted in yellow.

The previous Figure 4B is now 4A and vice versa. We labelled H261, Q315, E366, F292 and D265 in the surface presentation. As the color coding was potentially misleading between the two subfigures, we now colored bond forming residues in red instead of yellow in Fig. 4B.

- Fig. 5A: individual data points should be shown in the bar charts and is the error displayed SEM or SD?

We now display the individual data points in the bar diagram. The error is displayed as SD, as also in Appendix Fig. S2 and described in the figure captions.

- Fig. 5C: Leu264 and Asp265 should be included to illustrate why these have been chosen for mutagenesis.

In Figure 5C we intend to illustrate why a decrease in IC_{50} upon mutation of Q315 is seen for VSM, but not for AZ1 or FT206. The caption of Fig. 5C is therefore called: "Displacement of the Q315 side chain upon inhibitor binding". For the illustration of residue choice in our mutational studies and the binding site we complemented Fig 1C with L264 and extended the Appendix with a Ligplot figure (Appendix Fig. S1C) of all three complex structures in addition to Figures EV1C.

- Fig. 6A: label key structural elements in the figure.

This has now been included.

- A comparison with other USP-inhibitor structures, where the inhibitors also bind to the palm thumb cleft should be provided as a figure (superposition or side by side).

We introduced a new figure EV1B comprising a superposition of known USP7 inhibitor complex structures in which the inhibitors Cpd2 (PDB 5WHC) as well as GNE6776 (PDB 5UQX) in the thumb-palm cleft are shown. A third inhibitor (GNE6640, PDB 5UQV) is known to bind in a highly similar manner to GNE6776 and has therefore been omitted in this Figure. Additionally, we show 2 representative USP7 inhibitors (ALM2, PDB 5N9R and FT827, PDB 5NGF), which bind into the catalytic cleft. As these inhibitors stabilize the USP7 helix corresponding to USP28 α 5, in a position which deviates from the position, required to assume the Ub-binding competent state, we also included USP7 in its apo and Ub-bound states in the superposition. For easy comparison with the USP28 - inhibitor structures, this subfigure was placed below the corresponding subfigure which compares the AZ1, VSM and FT206 complex structures (EV1A).

- Fig EV2A: label residues highlighted in yellow; clearly label position of E366 and H261 in figures.

As also seen in the USP28 – Ub structure, where residues forming H-bonds with Ub are shown in yellow, H261 and E366 would be hardly visible, we have therefore decided to omit coloring of these two residues.

- Fig. EV5: label key sec. structure elements

This has now been included.

Referee #3:

This is a nice study describing X-ray co-structures of three chemically distinct dual USP25/28 inhibitors to USP28 accompanied by biophysical and biochemical studies using wild-type and mutant USP25 and USP28 enzymes to characterize key determinants of binding and inhibition as well as the effect of compound binding on enzyme dimers and tetramers. The data combine for an excellent story providing novel insight into USP25/28 inhibition as well as DUBs more generally. Studies are logical, rigorous and well described. The manuscript is recommended for publication following some minor edits/clarifications.

- Need to clearly define S1. In DUB space, this refers to distal ubiquitin binding site where as in general protease literature it refers to the binding cleft for the P1 amino acid of the substrate.

Thanks a lot for pointing this out. When we mention the S1 site for the first time in the results part of the revised manuscript we explain the meaning of the S1 site in the DUB space.

- The current text in both the results and discussion sections suggests that the USP1 inhibitor ML323 binds in the same cleft as the USP28 inhibitors and USP7 inhibitors. The USP1 inhibitor ML323 binds in a cryptic pocket on the other side of the DUB domain. This needs to be clear.

Thanks again for reading our manuscript so thoroughly. This statement was incorrect and we removed ML323/USP1 from the text. Instead, we now show an overview of other thumb-palm cleft and catalytic channel binding USP7 inhibitors in Fig. EV1B.

- Since USP25 exists in an autoinhibited state, it has been shown to be less active in biochemical assays than USP28. A comment on how this impacts comparing the IC50s for the two enzymes to each other will be useful.

In our experiments characterizing the IC_{50} values of the three compounds (Fig. 5) we utilized a construct of USP25, cat Δtip , which is purely dimeric and fully active due to the lack of the auto-inhibitory tip. It is correct that even the dimeric and active USP25 is still a less active enzyme than USP28 as previously described (Sauer *et al*, 2019; Gersch *et al*, 2019). We therefore mainly focused on IC_{50} comparisons within the same enzyme and its variants instead of comparisons between USP25 and USP28.

Dear Prof. Kisker,

Thank you for the submission of your revised manuscript to EMBO Reports. As I already informed you, we have received the report from the referee who was asked to assess it, who supports publication in EMBO Reports.

Browsing through the manuscript myself, I noticed a few editorial things that we need before we can proceed with the official acceptance of your study.

- Data availability section:

a) Please insert links that resolve directly to the datasets deposited at PDB.

b) The section is meant to refer to datasets deposited at public repositories only, therefore, please remove "Expression vectors will be provided upon reasonable request to the corresponding author Caroline Kisker."

c) In addition, the section needs to be placed before the Acknowledgements.

- Regarding the Author Contributions, we now use CRediT to specify the contributions of each author in the journal submission system. Therefore, please remove the Author Contributions from the manuscript file and make sure that the author contributions in our manuscript tracking system are correct and up-to-date. The information you specified in the system will be automatically retrieved and typeset into the article. You can enter additional information in the free text box provided, if you wish.

- All information on funding needs to be complete in the online manuscript tracking system, as this information will be transferred to our publisher and PubMed. In this respect, we note that the information on funding by the "Rudolf Virchow Center for Integrative and Translational Bioimaging" has been listed in the Acknowledgements section but not in the manuscript tracking system.

- Figure callouts: Table 1 needs to be "Table 1" throughout the manuscript in the text citations (there are instances of just "Table"). In addition, please change "[...] is described in the Supporting Information (Fig. EV5C)." to "[...] is described in (Fig. EV5C)." [page 18]

- Materials and Methods should be Methods

- Please describe your findings in the Abstract in present tense.

- Figure 5, legend: Please move the following sentence: "Data in A and B are presented as average {plus minus} SD. All the statistics show the results of unpaired Student's t tests, with p-values (* {less than or equal to} 0.05; ** {less than or equal to} 0.01; *** {less than or equal to} 0.001; **** {less than or equal to} 0.0001)" to the end of the legend and precede it with "Data Information: ...".

Preferably, the exact p-values are given, instead of the range.

- Please note that the error bars are not defined in the legend of Figure EV 4b.

- Source data need to be uploaded as one folder per figure, please.

- Data citations: Our journal encourages inclusion of *data citations in the reference list* to directly cite datasets that were re-used and obtained from public databases.

Data citations in the article text are distinct from normal bibliographical citations and should directly link to the database records from which the data can be accessed. In the main text, data citations are formatted as follows: "Data ref: Smith et al, 2001" or "Data ref: NCBI Sequence Read Archive PRJNA342805, 2017". In the Reference list, data citations must be labeled with "[DATASET]". A data reference must provide the database name, accession number/identifiers and a resolvable link to the landing page from which the data can be accessed at the end of the reference. Further instructions are available at <<https://www.embopress.org/page/journal/14693178/authorguide#referencesformat>>.

I noted the following instances where you seem to have compared your structural data to published structures (A - C, below). You would cite both, the original paper and the structure. For example, In the case of Gersch et al, 2019 you would cite (Gersch et al 2019; Data ref: Gersch et al, 2019) and in the reference list you first list the paper from Gersch et al, 2019 and then the dataset: Authors, year, database and name of dataset (link to dataset). [DATASET]

A) For comparison, the structure of USP28 Δ UCID (PDB 6HEI; grey, cartoon) in complex with Ub-PA bound to the S1-site (orange, α -trace) is shown (Gersch et al, 2019)

B) Crystal structure of the auto-inhibited USP25 catalytic domain (PDB 6H4J) (Sauer et al, 2019).

C) Superposition of structures of USP7 apo (PDB 1NB8), Ub-bound (PDB 1NBF) and in complex with the thumb-palm cleft binding inhibitors Cpd2 (PDB 5WHC; bright green), GNE6776 (PDB 5UQX; pink) and the catalytic channel binding inhibitors ALM2 (PDB 5N9R; dark green) and FT827 (PDB 5NGF; cyan) (Hu et al, 2002; Di Lello et al, 2017; Kategaya et al, 2017; Turnbull et al, 2017; Gavory et al, 2018).

- On a different note, I would like to alert you that EMBO Press offers a new format for a video-synopsis of work published with us, which essentially is a short, author-generated film explaining the core findings in hand drawings, and, as we believe, can be very useful to increase visibility of the work. This has proven to offer a nice opportunity for exposure i.p. for the first author(s) of the study. Please see the following link for representative examples and their integration into the article web page:

<https://www.embopress.org/doi/full/10.15252/embj.2019103932>

With kind regards,

Referee #2:

The authors have satisfactorily addressed all the points previously raised and I now recommend publication of the manuscript.

The authors have addressed all minor editorial requests.

Prof. Caroline Kisker
University of Wuerzburg
Rudolf Virchow Center for Integrative and Translational Bioimaging
Josef Schneider Strasse 2
Würzburg 97080
Germany

Dear Prof. Kisker,

I am very pleased to accept your manuscript for publication in the next available issue of EMBO reports. Thank you for your contribution to our journal.

Kind regards,
